# AUTOMATING BENCHMARK DESIGN

## ABSTRACT

The rapid progress and widespread deployment of LLMs and LLM-powered agents has outpaced our ability to evaluate them. Hand-crafted, static benchmarks are the primary tool for assessing model capabilities, but these quickly become saturated. In contrast, *dynamic benchmarks* evolve alongside the models they evaluate, but are expensive to create and continuously update. To address these challenges, we develop `BeTaL` (Benchmark Tuning with an LLM-in-the-loop), a framework that leverages environment design principles to ***automate the process of dynamic benchmark design***. `BeTaL` works by parameterizing key design choices in base benchmark templates and uses LLMs to reason through the resulting parameter space to obtain target properties (such as difficulty and realism) in a cost-efficient manner. We validate this approach on its ability to create benchmarks with desired difficulty levels. Using `BeTaL`, we create two new benchmarks and extend a popular agentic benchmark $\tau$-bench. Extensive evaluation on these three tasks and multiple target difficulty levels shows that `BeTaL` produces benchmarks much closer to the desired difficulty, with average deviations ranging from 5.3% to 13.2% — a 2-4$\times$ improvement over the baselines.

## 1 INTRODUCTION

New developments in LLMs, particularly in powering agents via advanced planning, reasoning, and tool-use capabilities (Valmeekam et al., 2023; 2024; Ferrag et al., 2025), have outpaced current methods for evaluation. Static, human-curated benchmarks, such as GPQA (Rein et al., 2024) or HLE (Phan et al., 2025), remain popular, but are costly to develop and quickly become obsolete as models continue to improve. This is challenging for model developers, as increasingly saturated benchmarks make it impossible to differentiate between the performance of state-of-the-art models.

To address these challenges, researchers have turned to **dynamic benchmarks** that can be updated over time. These benchmarks avoid saturation via re-calibration or the introduction of new and harder data; this also limits the risk of train-test *contamination*. For example, LiveBench (White et al., 2024) periodically introduces new questions and harder tasks. However, these types of benchmarks still largely rely on *unscalable human authoring and manual updates*. Increasingly popular agentic tasks exacerbate this problem, as simulated environments must be carefully crafted; repeatedly designing and implementing new environments promises to be even more labor-intensive.

How can we build dynamic benchmarks for frontier LLMs without the expense and inefficiency of ongoing manual design and implementation? Unsupervised environment design (UED) methods (Jiang et al., 2021a) work with environments that are built from abstract task templates with a set of configurable parameters. These parameters can be tuned to produce new and higher utility versions of the benchmark, thus enabling dynamic re-use. In practice, however, we find that the search space over such parameters is intractable for non-trivial environments. Naively sampling random configurations is inefficient, as many will be trivial or unsolvable.

We overcome these obstacles via a new approach, **Be**nchmark **T**uning with **a**n LLM-in-the-loop (BeTaL), that performs dynamic benchmark design. `BeTaL` leverages the capabilities of large reasoning models playing the role of designers. Central to our approach is the use of a powerful ***designer LLM*** tasked with reasoning over the space of possible parameter values, design choices, or tasks. The designer is prompted to consider the various parameters of an under-specified benchmark or environment and to propose instances or values that are expected to be high utility. This is set up as an interactive and iterative process: after the designer has specified an environment, a simulator

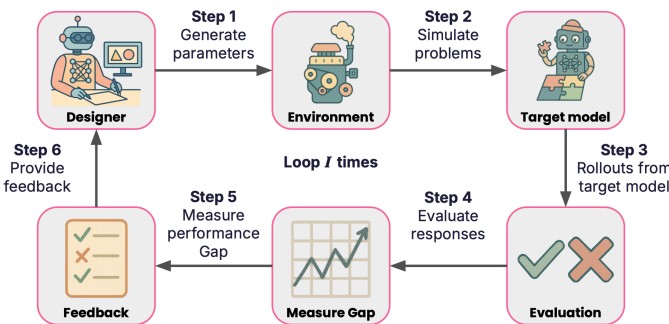

Figure 1: `BeTaL` automates the process of designing and adjusting *dynamic benchmarks* to meet target criteria.

creates a sample benchmark (problems with ground truth answers), and the model or agent being evaluated attempts this benchmark, with results provided back to the designer. After each round, the designer must reason over choices and results and make *changes in the parameter values so that the new parameters will result in a benchmark with desired objectives (such as difficulty or realism)*. This closed-loop multi-round strategy allows the benchmark to dynamically adjust over time to meet the objectives. While the procedure is flexible to incorporate several types of objectives and combinations thereof, here we focus on the objective of creating a benchmark with a given target difficulty level.

We hypothesize that the strong zero-shot or few-shot reasoning capabilities of frontier models enable the designer to understand the factors that influence usefulness (e.g., task difficulty) and design benchmarks that meet all desirable criteria (e.g., tasks that are just outside of a weaker model's current capabilities). This framework design reduces the burden of designing and continually updating benchmarks to meet the demands of ever-improving models. In addition, it permits re-purposing of existing static benchmarks - breathing new life into datasets long considered outdated.

Our contributions are:

- **A flexible framework for automating benchmark design.** We posit that benchmark properties such as complexity are determined by a set of underlying benchmark parameters. With this insight, we formulate the design process as an optimization problem over the space of benchmark parameters to obtain settings that will result in a benchmark having desired properties, e.g., difficulty level.

- **An efficient LLM-based procedure to solve the optimization.** Leveraging the reasoning capabilities of frontier large language models, we introduce Benchmark Tuning with an LLM-in-the-loop (`BeTaL`) to efficiently solve the above optimization problem for benchmark design.

- **New benchmarks and empirical validation**: Using `BeTaL`, we modify existing benchmarks to meet new requirements for dataset-level difficulty, and we introduce new benchmarks that focus on mathematical and spatial reasoning. Our extensive empirical evaluation of `BeTaL` on these settings reveals `BeTaL` consistently obtains benchmarks with low deviation (5.3% - 13.2%) between observed and target difficulty — a 2-4$\times$ improvement over baselines across all tasks.

## 2 METHODOLOGY

We propose `BeTaL`, a novel framework that uses an LLM-in-the-loop to iteratively design dynamic benchmarks that achieve user-specified goals. Before describing the algorithm, we outline the key building blocks of the system.

---

**Algorithm 1** Benchmark Tuning with an LLM-in-the-loop (`BeTaL`)

---

1: **Input:** Under-specified Environment Description, Parameter Set $P$, Target Performance $\rho$, Target Model $M_t$, Designer Model $M_d$, Number of Iterations $I$.
2: Initialize $i^* \leftarrow 0$, $v_{i^*} \leftarrow \emptyset$, minimum gap $\hat{g}_{i^*} \leftarrow \infty$
3: **for** $i = 1$ to $I$ **do**
4:     `Prompt` $\leftarrow$ Template with Environment description, $P$, $\rho$
5:     **if** $i > 1$ **then**
6:         `Prompt` $\leftarrow$ `Prompt` + Summary of previous iterations
7:     **end if**
8:     $v_i \leftarrow M_D(\texttt{Prompt})$                         ▷ Get parameters from Designer Model
9:     $v_i \leftarrow \text{ProjectToDomain}(v_i, \mathcal{V})$
10:    $D_i \leftarrow \text{InstantiateSimulator}(v_i)$          ▷ Generate problems with simulator
11:    $\hat{\rho}_i \leftarrow \text{EvaluateModel}(M_T, D_i)$            ▷ Evaluate Target Model
12:    $\hat{g}_i \leftarrow |\hat{\rho}_i - \rho|$
13:    Update summary of previous iterations with $v_i$ and $\hat{\rho}_i$   ▷ Step 4: Prepare feedback for next iteration
14:    **if** $\hat{g}_i < \hat{g}_{i^*}$ **then**
15:       $i^* \leftarrow i$
16:       $\hat{g}_{i^*} \leftarrow \hat{g}_i$
17:    **end if**
18: **end for**
19: **Return:** $v_{i^*}$

---

## 2.1 PRELIMINARIES

Our approach assumes a loosely defined environment template that can be refined and instantiated into concrete benchmarks. The system consists of the following components:

**Underspecified environment.** The user begins with a high-level description of the benchmark they want to create—for example, a spatial reasoning benchmark where questions involve tracking objects on a grid after a sequence of transformations. Intuitively, the complexity of problems depends on several factors such as grid size, number of actions, types of operations, etc. We begin with an underspecified environment, wherein the environment is characterized by a finite set controllable parameters $P = \{p_1, p_2, \ldots, p_k\}$, $p_i \in V_i$, so that the overall design space is $\mathcal{V} = V_1 \times V_2 \times \ldots \times V_k$.

**Problem/Task Generator.** We assume access to a simulator that, given a parameter configuration $v \in V$, can instantiate the environment and generate a dataset $D = \{(x_j, y_j)\}$ of problems with ground-truth solutions. It is expected that the simulated problems adhere to the constraints specified by the parameter values. In this work, we focus on environments with verifiable or procedurally generated solutions, allowing us to assume that the generated ground truth is correct.

**Target model.** A model or system to be evaluated, e.g., an off-the-shelf LLM, a proprietary API, or a multi-agent pipeline.

**Target performance.** Along with the target model, the user also specifies a target performance level $\rho \in \mathbb{R}$ and a distance measure $d$. The objective is to output a benchmark on which the target model's performance will be close to $\rho$. The exact definition of $\rho$ is left to the user; for instance $\rho$ could be accuracy, diversity, or an aggregate of multiple measures. In this work, we use target difficulty as our measure of performance of the generated benchmarks, as we seek to overcome the challenge of benchmark saturation.

**Designer model.** A sufficiently powerful model, such as large reasoning models (LRMs), that can understand the underspecified environment description, the set of free parameters and constraints that influence the environment's complexity. We expect such a model to be able to reason about the design space and propose specific values to the parameters that result in an environment of given target complexity.

## 2.2 BeTaL: Benchmark Tuning with LLM-in-the-loop

BeTaL is built on two key ideas: first, strengthening grounding through explicit feedback from real rollouts of the designed benchmarks; and second, leveraging LLM reasoning to systematically explore and refine the design space. This process mirrors how humans design benchmarks; through an iterative loop of experimentation and observation, where both elements are essential for effective benchmark creation. We describe the process in Alg. 1, and explain it in detail below.

**Step 1: Parameter generation (LLM-Guided).** In step one of the BeTaL, the designer model, an LRM, is prompted to obtain a parameter configuration $v_i$. Since these values are generated by a language model, they may be out of the domain $\mathcal{V}$. We found designer models occasionally hallucinated out of domain configurations in roughly 4% of proposals. Verification is therefore necessary to ascertain that $v_i \in \mathcal{V}$, and, if not, this process is repeated until the generated $v_i$ falls in $\mathcal{V}$. In the end, $v_i$ is projected to $\mathcal{V}$ if it is still out-of-domain.

**Step 2: Environment instantiation and problem/task generation.** A simulator is instantiated with the parameter configuration obtained in Step 1, which is then used to generate a small set of problems/tasks, with ground truth answers for evaluation, i.e. $D_i = \{(x_j, y_j)\}_{j=1}^{n_s}$.

**Step 3: Performance evaluation.** The target model is evaluated on $D_i$ to yield performance $\hat{\rho}_i$. When the ground truth is not available, $\hat{\rho}_i$ could be estimated by evaluating using LLM-as-a-Judge (Gu et al., 2024) or Program-as-a-Judge (Huang et al., 2025a).

**Step 4: Feedback and iteration.** The iteration details, including the parameter choices and the resulting performance, are summarized in natural language to the LRM, including $v_i$ and $\hat{\rho}_i$. This feedback is appended to the next prompt, enabling the model to reason about the impact of its prior choices and propose improved parameters in subsequent iterations.

**Step 5: Termination and selection.** In each iteration, we keep track of the observed performance gap $\hat{g}_i = |\hat{\rho}_i - \rho|$ and keep track of the iteration $i^*$ that results in the smallest gap. After $I$ iterations, the method exits and returns $v_{i^*}$.

## 3 Experimental Setup

In this section, we describe our setup for the experiments. First, we give high-level details of the benchmarking tasks, then discuss the baseline methods, our choices of designer and target models, evaluation metrics, and the protocol to run the experiments.

### 3.1 Benchmarking Tasks

We consider a range of tasks based on arithmetic, spatial reasoning, and airline customer service agents. Each of these settings has a rich design space with several free parameters that govern the complexity of the benchmark, making them good candidates for evaluating our method. We briefly discuss these tasks and defer the details to the Appendix A.1.

**Arithmetic sequences task.** Given an input number $x \in \mathbb{R}$ and an output number $y \in \mathbb{R}$, an agent must return the sequence of arithmetic operations $o_1, o_2 \ldots o_N$ that, when applied recursively to the intermediate results, yield $y := (o_N \circ o_{N-1} \circ \cdots \circ o_1)(x)$. At inference time, the target model, an LLM agent, is provided access to the arithmetic operators, as tools, to determine the sequence of operators that transform $x$ to $y$. The predicted operator sequence is verified by executing the sequence and comparing it with the ground truth $y$. Task difficulty depends on several factors such as operator choice, sequence length, range of the input $x$, and others.

**Spatial reasoning task.** We design multiple spatial reasoning tasks involving a 2D square grid (board) with particles placed on it. The board and particles can both rotate, while the particles can additionally move positions. A series of such actions is applied, after which the model is queried about the final positions and orientations of the particles. The target LLM receives a description of the environment and action sequence, and its responses are compared against programmatically computed ground truth. The complexity is controlled by parameters such as board size, the number and types of actions allowed.

$\tau$-**bench "airline" task.** This is an interactive evaluation environment for customer service agents in simulated airline scenarios, where the agent must use available tools to query and update a database to fulfill user requests (Yao et al., 2024). The reward is computed by comparing the final database state with the database state following a series of golden actions. Building on this setup, we design a rule-based task generator that randomly samples action sequences and corresponding user instructions. The generator is parameterized both by tool-related variables—such as the number of passengers when booking a flight—and by behavioral parameters derived from real user instructions.

On all three problems, our objective is to identify parameter configurations that yield benchmarks with desired difficulty levels. For further details on these tasks and associated parameters, see Appendix A.1.

### 3.2 BASELINES

We briefly discuss the baselines for evaluation. Details are provided in Appendix A.2.

**Random sampling with prioritized parameter replay (RS+PPR).** Inspired by Prioritized Level Replay (PLR) (Jiang et al., 2021b), we develop a baseline RS+PPR, that maintains a buffer of favorable environment parameters. In each iteration, it samples a parameter configuration $v_i \in \mathcal{V}$ either uniformly at random (with probability $p$) or, with probability $1 - p$, as a noisy variant of parameters drawn from the buffer. Then the performance gap $\hat{g}_i$ is estimated with $v_i$, and it is added to the buffer if $\hat{g}_i \leq \Delta$.

**Best-of-N variations.** We use best-of-$N$ (BoN) Snell et al. (2024); Beirami et al. (2025), where $N$ responses are sampled and the best response selected according to a reward model. We consider the reward for a parameter configuration to be the negative of its observed performance gap. In the first variant, we consider **BoN-ML**, with our verifier as a predictive model trained offline using standard machine learning methods on parameter–performance-gap pairs. In the second variant, **BoN-TM**, we collect a small number of rollouts with the target model, and select the response with the smallest measured performance gap.

### 3.3 DESIGNER AND TARGET MODELS

We use the latest reasoning models: GPT-5, Claude Opus 4.1, and Grok 4 as designer models and o4-mini as the target model in all settings. We evaluate the resulting benchmarks on three models: o4-mini, Gemini 2.5 Flash, and Claude 3.7 Sonnet. Whenever applicable, we configure the designer model with temperature 0.5 and a reasoning budget of 4096 tokens for exploration, while the other models use temperature 0.0 with a reasoning budget of 1024 tokens. Details of model configurations are in Appendix A.3.

### 3.4 METRICS

Each benchmarking task can have its own notion of performance $\rho$ (e.g., accuracy, pass@k, etc.). We assume this measure is inversely proportional to the task difficulty, and define the following metric:

**Performance gap.** If a method is run with a given target performance level $\rho$, and say that it results in a benchmark on which the target model has performance $\hat{\rho}$, then its performance gap is $\hat{g} = |\hat{\rho} - \rho|$.

### 3.5 EXPERIMENT PROTOCOL

We evaluate the methods across two phases: parameter search and evaluation. During parameter search, iterative methods are run for 10 iterations, while non-iterative methods sample 10 configurations. The best parameters obtained from each method are then used to generate a larger evaluation dataset. To assess each designer's ability to produce benchmarks with controlled difficulty, we define four target performance levels: Hard ($\rho^{\text{hard}} = 0.25$), Medium ($\rho^{\text{medium}} = 0.50$), Easy ($\rho^{\text{easy}} = 0.75$), and Trivial ($\rho^{\text{trivial}} = 0.90$). The primary evaluation metric is the average performance gap, $\bar{\hat{g}}$, computed at each level. All experiments are repeated three times with different random seeds, and results are reported with 95% confidence intervals based on the Student's-t distribution with three degrees of freedom.

Table 1: `BeTaL` consistently outperforms the iterative and Best-of-N baselines in both parameter search and evaluation phases across all three tasks and all three designer models. Reported numbers are $\bar{\bar{g}}(\%)$ with o4-mini as the target model. For parameter search, we run either 10 samples or 10 iterations and report the best result for a fair comparison. More experimental details can be found in Appendix A.

| Designer | Method | Arith. Seq. | | Spatial Reasoning | | $\tau$-Bench Airline | |
| --- | --- | --- | --- | --- | --- | --- | --- |
| | | Param Search | Eval | Param Search | Eval | Param Search | Eval |
| N/A | RS+PPR | 15.8±2.43 | 13.11 ±11.6 | 6.6±12.7 | 8.36±10.45 | 18.3±21 | 21.3±10.6 |
| GPT-5 | BoN-TM | 8.3±4.64 | 11.67±7.67 | 28.34±41.19 | 30.26±42.77 | 12.5±2.1 | 20.8±8.0 |
| | BoN-ML | 30.0±12.63 | 22.17± 17.99 | 21.66±19.67 | 31.20±43.49 | 21.4±11.5 | 16.7±10.4 |
| | BeTaL | **5.8±4.77** | **9.0± 8.49** | **0.4±0.35** | **5.34±12.77** | **5.3±3.2** | **13.2±10.3** |
| Opus-4.1 | BoN-TM | 20.0±12.12 | 18.94± 18.72 | 26.93±41.32 | 31.07±43.27 | **3.6±3.2** | 10.0±12.4 |
| | BoN-ML | 31.7±6.20 | 29.17± 6.06 | 20.49±19.13 | 32.76±43.76 | 11.7±7.5 | 9.7±7.6 |
| | BeTaL | **12.5±4.42** | **11.78±10.5** | **3.82±5.58** | **7.35±5.49** | 5.0±2.1 | **7.7±5.2** |
| GROK 4 | BoN-TM | 20.0±11.70 | 21.44±11.05 | 25.36±39.60 | 29.76±43.64 | 15.0±11.5 | 18.5±7.7 |
| | BoN-ML | 32.5±15.58 | 33.11±20.22 | 21.24±19.44 | 33.81±46.05 | 34.2±14.3 | 20.2±3.1 |
| | BeTaL | **4.2±3.26** | **8.28 ± 4.30** | **1.36±2.72** | **4.98±8.13** | **3.9±3.2** | **10.3±12.4** |
| Designer Avg. | BoN-TM | 16.11±6.16 | 17.35±6.03 | 26.88±40.70 | 30.36±43.23 | 10.37±4.17 | 16.4±4.89 |
| | BoN-ML | 31.39±8.89 | 26.81±8.22 | 21.13±19.41 | 32.59±44.44 | 22.41±7.88 | 15.5±4.67 |
| | BeTaL | **7.50±2.57** | **9.69±2.93** | **1.86±2.88** | **5.89±8.80** | **4.72±1.86** | **10.39±3.18** |

Table 2: Chain-of-thought (CoT) prompting does not consistently yield strong designer-model performance. While Claude Opus-4.1 achieves competitive results on the arithmetic sequence and $\tau$-Bench tasks, state-of-the-art LLMs often struggle to outperform a random sampling baseline. Reported values are $\bar{\bar{g}}(\%)$ with o4-mini as the target model.

| Method | Arith. Seq. | Spatial Reasoning | $\tau$-Bench Airline |
| --- | --- | --- | --- |
| Random Sampling | 21.17±51.5 | 25.4± 9.6 | 37.3±17.2 |
| CoT Prompting (GPT-5) | 28.33±25.8 | 45.3± 26.3 | 23.6±16.1 |
| CoT Prompting (Opus-4.1) | 11.67±7.2 | 26.1± 17.9 | 11.9±10.4 |
| CoT Prompting (Grok-4) | 20.83±3.6 | 39.1± 25.5 | 31.9±13.3 |

## 4 RESULTS AND DISCUSSION

In this section, we present our main results and discussion. We provide an in-depth discussion on `BeTaL`'s effectiveness in designing benchmarks for any given target difficulty.

**C1: `BeTaL` outperforms baselines in creating benchmarks with any target performance level.**

Our hypothesis is that while LLMs are highly capable, a single round of prompting, even with a large reasoning budget, is less effective than an iterative framework like `BeTaL`, which incorporates feedback from previous rounds. Drawing inspiration from recent work framing LLMs as optimizers (Yang et al., 2023; Nie et al., 2024), we expect `BeTaL`'s feedback-driven search to yield stronger performance than non-iterative baselines. The results in Table 1 strongly support this hypothesis. We summarize the key findings below.

*i) `BeTaL` versus other multi-round methods.* We compare `BeTaL` with multi-round baselines, including RS+PPR and the variations of Best-of-N. From our results (Table 1), it is evident that `BeTaL` outperforms these baselines by a wide margin, across benchmarks and designer models. We attribute this advantage to the reasoning capacity of LLM-based designers, which enables them to iteratively refine parameters using feedback from previous rounds. In contrast, other baselines, including those that receive feedback, fail to exploit it as effectively. `BeTaL`'s capabilities in iteratively finding the target parameters can be further seen in Figure 3 and Figure 11 in the Appendix.

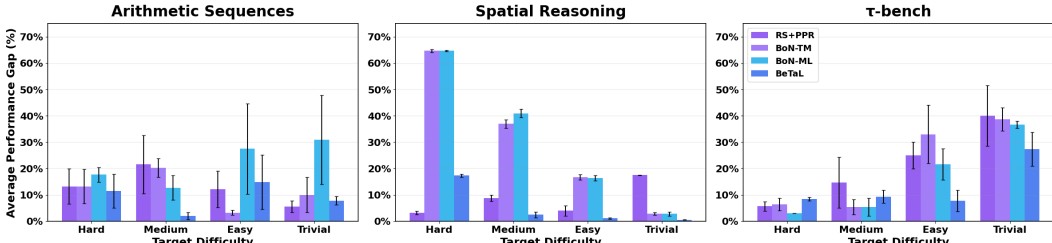

Figure 2: Evaluation results on o4-mini with `BeTaL` (with GPT-5 as the designer model, and o4-mini as the target model during parameter search) perform robustly at different target difficulty levels, compared to baselines on Arithmetic Sequences, Spatial Reasoning, and $\tau$-Bench. A similar performance is noted using Claude Opus 4.1 and Grok-4 as Designers, in Figure 9 in the Appendix.

It shows that `BeTaL` shrinks the performance gap more strongly than `RS+PPR` over 10 iterations, with a wide margin (more than 20%) on both $\tau$-Bench and Spatial Reasoning.

***ii) Performance at target difficulty levels.***

We expect an effective benchmark designer to optimize for any specified target difficulty level. Figure 2 presents the observed performance gap for each target difficulty level. `BeTaL` demonstrates strong robustness, consistently outperforming all baselines at each difficulty level.

We also observe inherent difficulty differences across benchmark domains, which are reflected in the performance gaps. For example, $\tau$-Bench and Spatial Reasoning are inherently challenging, with the largest gaps appearing at the Trivial difficulty level for all LLM designers. In contrast, the Arithmetic Sequence task, containing several degenerate solutions, shows the largest gap at the Hard difficulty level (see Figure 10 in the Appendix).

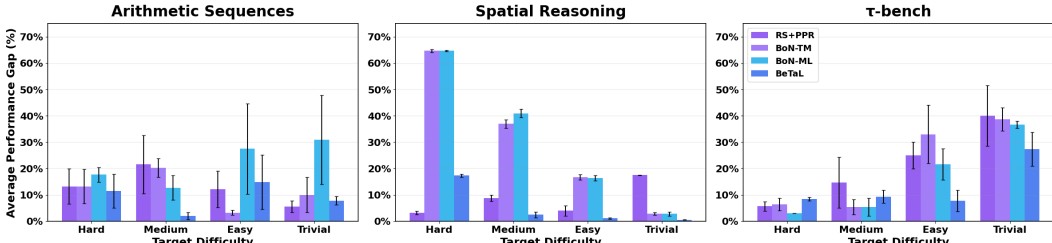

Figure 3: Convergence of iterative methods during parameter selection on Spatial Reasoning and $\tau$-Bench benchmarks: `BeTaL` vs. `RS+PPR`. Performance gap of `BeTaL` shrinks faster compared to `RS+PPR`, within 10 iterations, indicating LLMs are more efficient than competing iterative methods at finding favorable environment parameters for benchmark creation. Results are averaged over difficulty levels and designer models.

***iii) Performance comparison of designer models.*** While `BeTaL` achieves strong performance with all three designer (reasoning) models, we find that the choice of reasoning model may depend on the nature of the benchmark being developed. Comparing between the designers, Grok-4 and GPT-5 do well on the mathematical and logical reasoning domains of Arithmetic Sequences and Spatial Reasoning. On the other hand, Claude-Opus-4.1 excels on the real-world agentic benchmark of $\tau$-Bench Airline, with a performance gap of $7.7 \pm 5.2\%$ compared to $13.2 \pm 10.3\%$ and $10.3 \pm 12.4\%$ by GPT-5 and Grok-4, respectively (Table 2).

**C2: Benchmark created by `BeTaL` for one target model is transferable to other target models.**

A benchmark designed for a target model (here, o4-mini) can also be used to evaluate other models. When the target and evaluation models coincide, `BeTaL` produces benchmarks with minimal performance gaps. However, when evaluated on models different from the target, performance naturally varies with model capability. For instance, a benchmark that is hard for the target model may appear of medium difficulty to a stronger model, and vice versa. Consequently, models with similar capabilities to the target are expected to exhibit comparable performance gaps, whereas stronger (or weaker) models should follow the same performance trends across target difficulty levels but with larger (or smaller) magnitudes.

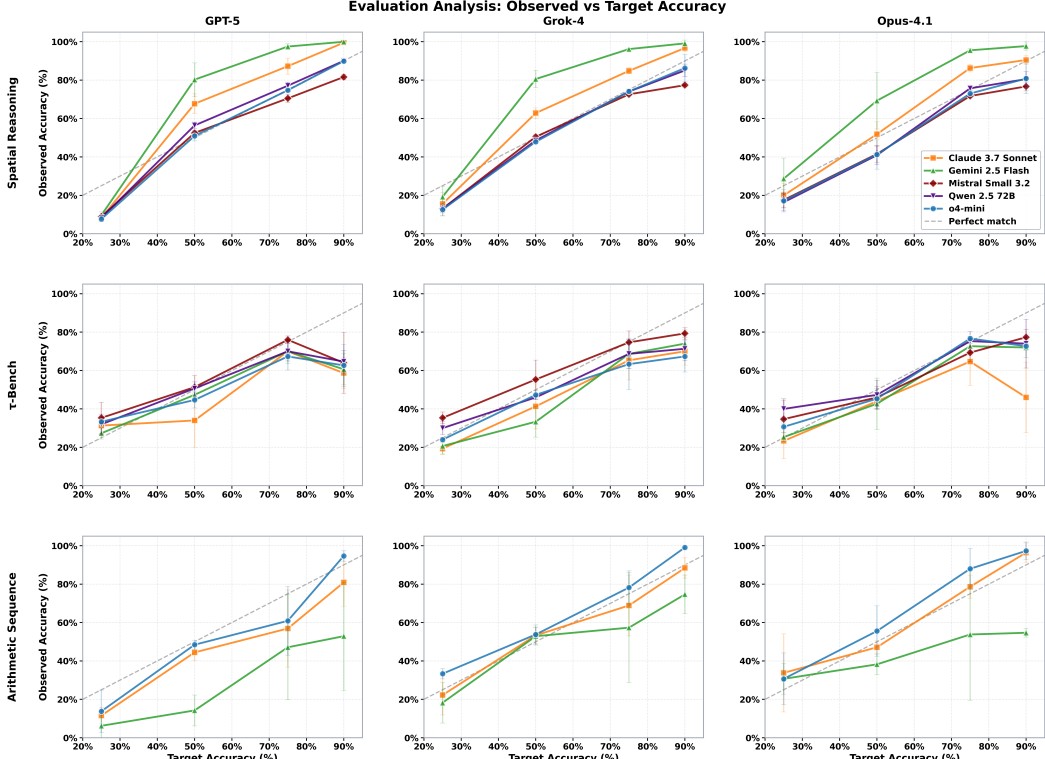

Figure 4: Evaluation generalization across designer models and datasets. Colored lines represent individual model eval performance (see legend for models) with respect to target accuracy. Observed versus target accuracy for o4-mini target trained by different designers (columns: GPT-5, Grok-4, Opus-4.1) on three benchmarks (rows: Spatial Reasoning, $\tau$-Bench, Arithmetic Sequence). The black dashed line indicates perfect alignment.

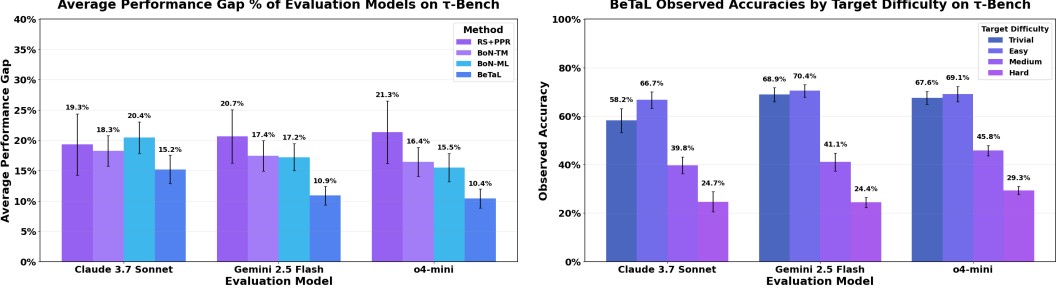

(a) Results averaged over the difficulty levels.

(b) BeTaL results at different target difficulty levels.

Figure 5: Results on different evaluation models. The left figure shows aggregate results for all methods, and the right figure focuses on BeTaL's results, showing the observed accuracies at different target difficulty levels. All results are averaged across Designer Models.

Our results in Figure 4 and 5 and confirm that benchmarks designed by BeTaL exhibit robust transferability across evaluation models. On $\tau$-Bench, benchmarks generated using o4-mini feedback yield comparable performance when evaluated on Claude 3.7 Sonnet and Gemini 2.5 Flash, with BeTaL consistently outperforming all baselines across evaluation models.

This cross-model consistency across different benchmark domains: agentic planning in real-world tasks ($\tau$-Bench) and mathematical reasoning (Arithmetic Sequences) domains provides strong evidence that BeTaL-designed environments test fundamental cognitive capabilities that generalize across different model architectures and families, rather than exploiting model-specific weaknesses.

**C3: Chain-of-Thought alone is insufficient for efficient benchmark design.**

Despite the remarkable reasoning capacity and extensive world knowledge of state-of-the-art LLMs, their ability to systematically design benchmarks, using prompting alone, remains unreliable. As shown in Table 2, even with high reasoning budgets, LLMs exhibit *high variance* when tasked with producing benchmarks of varying complexity. Using o4-mini as the target model, Claude Opus-4.1 surpasses the random baseline only on Arithmetic Sequence and $\tau$-Bench, but fails on Spatial Reasoning. GPT-5 and GROK 4 underperform even further. These results demonstrate that Chain-of-Thought prompting alone does not endow LLMs with robust or generalizable benchmark design capabilities.

**C4. Can LLMs also generate better parameter spaces?**

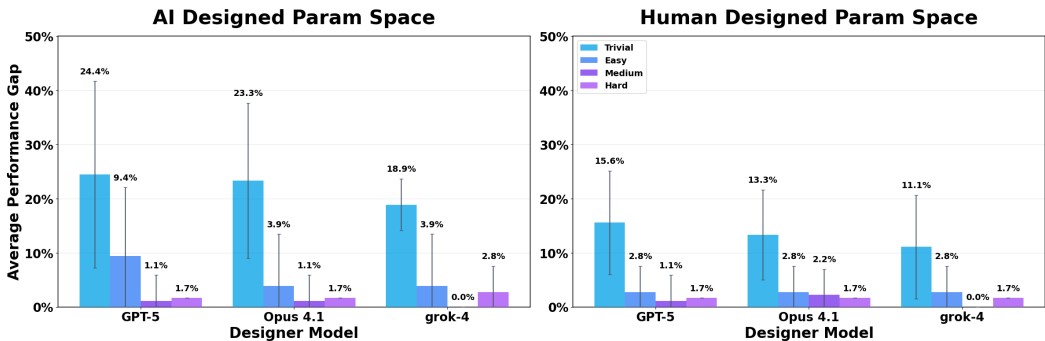

Figure 6: Performance of BeTaL on $\tau$-bench parameter space generated by Opus 4.1 versus by human. BeTaL on AI-generated parameter space is an acceptably small performance gap for medium and hard benchmarks, yet still generally underperforms to that generated by humans.

Given LLMs' strong ability to generate complex and diverse benchmarks through BeTaL, a natural question is whether they can also design the underlying *parameter spaces* themselves. To test this, we prompt Claude Opus-4.1, the best performing designer model on $\tau$-Bench, to generate a complete parameter space for $\tau$-Bench, then manually implement the feasible parameters in the task generator. Opus 4.1 adds additional parameters based on user interactions to the design space – including cooperation level, and clarifying preferences (whether explicit or implicit). Detailed parameters and prompts can be seen in Appendix A.1 and Appendix B.

As shown in Figure 6, BeTaL applied to the AI-generated parameter space performs comparably well on *Medium* and *Hard* benchmarks, achieving $\hat{g}$ as low as 1.1% and 1.7%, respectively. This demonstrates that LLMs can capture key structural patterns needed to produce challenging and well-calibrated benchmarks. However, a substantial gap remains relative to human-designed parameter spaces on *Trivial* and *Easy* benchmarks, reaching up to 24.4% and 23.3% performance gaps for GPT-5 and Opus-4.1, compared to 15.6% and 13.3% from the human-generated space. These gaps indicate limited flexibility and controllability in the LLM-generated parameter space, particularly in achieving smooth difficulty scaling across the full range of target performances.

Overall, these findings suggest that while current LLMs exhibit partial autonomy in environment design, fully self-sufficient parameter-space generation remains an open challenge for future systems.

## 5 RELATED WORK

**Automating benchmark design.** Recent work streamlines benchmark creation by automating generation, verification, and evolution. AUTOBENCHER (Li et al., 2025) introduces a declarative framework that automates benchmark construction by optimizing over benchmark desiderata to scalably discover new capability and safety weaknesses in language models. BENCHMAKER (Yuan et al., 2025) and CHASE (Patel et al., 2025) leverage LLMs for systematic or compositional task construction, with BENCHMAKER emphasizing structured evaluation and CHASE building harder problems from simpler components. In the code domain, graph-based generators validate solutions via loop-derived self-consistency and help train reliable LLM-as-judge proxies (Farchi et al., 2024).

Other approaches extend beyond static generation: tasks can evolve through perturbation, probing, or alternation (Wang et al., 2024), and multi-agent frameworks coordinate specialized roles for diverse benchmark creation (Butt et al., 2024). Despite this progress, most methods operate directly at the task level—fixing difficulty or other heuristics to guide evolution—without abstracting the environment design space that underlies task instantiation. This makes it hard to adapt benchmarks across new domains. Our approach instead parameterizes the benchmark and closes the loop with target model feedback, enabling flexible benchmark tuning.

**Environment design for curriculum learning.** Automated benchmark design parallels Unsupervised Environment Design (UED) in reinforcement learning, where tasks must remain solvable yet challenging as agents improve. UED methods adapt environments through adversarial generation (Dennis et al., 2021), replay-based curation (Jiang et al., 2021b), or evolutionary mutation (Parker-Holder et al., 2023). These approaches formalize environment design as optimization or curation to sustain adaptive curricula. Extending this idea, LLM-driven variants such as EnvGen (Zala et al., 2024) and LLM-POET (Aki et al., 2024) employ language models to generate or mutate RL environments, while co-evolutionary loops like R-Zero (Huang et al., 2025b) pair a Challenger and Solver in an adversarial, self-improving curriculum on language tasks. Although these methods share the goal of adapting difficulty in step with capability, BeTaL avoids the need for a training loop, enabling adaptive benchmark generation with open and closed models alike.

**Scaling environments and datasets.** A complementary line of work scales environments and datasets to advance agentic intelligence, often through synthetic generation or curated annotations. AgentScaler (Fang et al., 2025) builds large collections of verifiable, API-derived environments to train function-calling agents, while APIGen (Liu et al., 2024) and ToolACE (Liu et al., 2025) synthesize diverse, verifiable function-calling datasets through automated generation and multi-stage verification. More recently, ARE and its Gaia2 benchmark (Andrews et al., 2025) provide scalable, asynchronous environments that test adaptability and robustness. These efforts emphasize agentic capabilities, whereas our focus is on automating evaluation.

**LLMs as optimizers.** Our work fundamentally treats benchmark design as an optimization problem, with reasoning models as optimizers. Similar work has been explored in OPRO (Yang et al., 2024) and evolutionary variants such as LEO (Brahmachary et al., 2024) and Guo et al. (2025) to solve mathematical tasks and optimize prompts. Our work uniquely applies to benchmark design.

## 6 CONCLUSION, LIMITATIONS AND FUTURE WORK

We introduced BeTaL, an LLM-in-the-loop framework for dynamic benchmark design. Unlike static or manually maintained live benchmarks, BeTaL adaptively generates benchmarks that evolve with model capabilities. By reasoning over parameterized design spaces, it efficiently achieves target performance levels with minimal human input. Across arithmetic, spatial reasoning, and agentic domains, BeTaL consistently reduces performance gaps by 2-4× compared to LLM and non-LLM baselines. These results highlight BeTaL's potential to enable evaluation systems that evolve alongside advancing models. In light of BeTaL's adaptive and targeted task-generation capabilities, we note that its underlying ideas naturally relate to curriculum learning and could potentially inform curriculum based training strategies.

One of the drawbacks of BeTaL is that it assumes access to parameterized and verifiable task generators, which may not always exist. Its effectiveness depends on the reasoning strength of the designer model and careful prompt construction. Moreover, our evaluation is limited to a small set of domains, leaving multimodal and more subjective tasks unexplored. Although BeTaL is an evaluation based benchmarking framework rather than a training method, it appears structurally compatible with curriculum learning based training, but confirming this empirically is left for future work.

Future work could extend BeTaL to optimize multiple objectives, including realism and diversity, explore multi-agent or co-evolutionary design loops, and incorporate human-in-the-loop oversight to further enhance adaptability and reliability. Given its adaptive design, BeTaL also provides a promising basis for exploring *curriculum-based data selection* strategies. Ultimately, we envision adaptive benchmarks that evolve with the systems they evaluate, ensuring robust and meaningful assessment as AI capabilities advance.

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

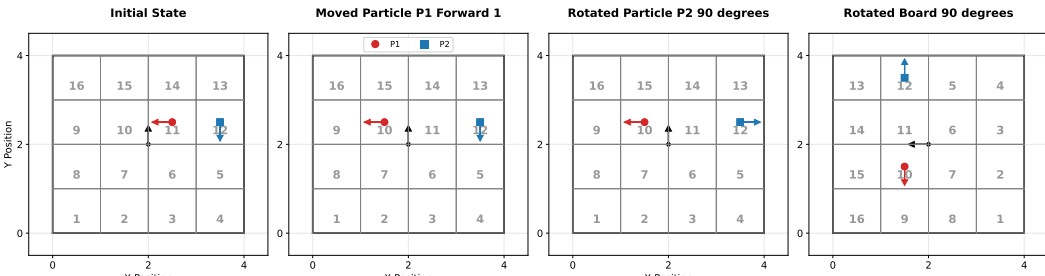

Figure 7: Illustration of particles and actions in spatial reasoning tasks. Here the board is 4x4 and initially oriented towards north (black arrow). There are two particles $P1$ and $P2$ oriented towards west and south respectively. The first action moved the particle $P1$ forward by one step, second action rotated the particle $P2$ by 90 degrees and the last action shows rotation of the board by 90 degrees. The board rotations are w.r.t. to its center and when a board rotates or moves the particles on it also rotate and move along with it.

# A    ADDITIONAL EXPERIMENTS AND DETAILS

## A.1    DETAILS OF BENCHMARKING TASKS

**Arithmetic sequences task.** Given an input number $x \in \mathbb{R}$ and an output number $y \in \mathbb{R}$, an agent must return the sequence of arithmetic operations $o_1, o_2 \ldots o_N$ that, when applied recursively to the intermediate results, yield $y$; i.e,

$$y = (o_N \circ o_{N-1} \circ \cdots \circ o_1)(x).$$

The benchmark space is constrained to simple operations of addition ($+$), subtraction ($-$), multiplication ($\times$), division ($\div$), square root ($\sqrt{\ }$), and power of two ($(\cdot)^2$). For binary operators, both operands are the same. At inference time, the target model, an LLM agent, has access to the arithmetic operators, as tools, to determine the sequence of operators that transform $x$ to $y$. The predicted operator sequence $o'_N, o'_{N-1}, \ldots, o'_1$ is verified by executing the sequence to generate

$$y' = (o'_N \circ o'_{N-1} \circ \cdots \circ o'_1)(x),$$

and comparing it with the ground truth $y$.

Task difficulty depends on factors such as operator choice, sequence length, range of the input $x$, and whether $x$ is integer or floating-point. Operators like subtraction or division tend to collapse $y$ toward zero, whereas multiplication and exponentiation operators cause exponential growth. Our automated benchmark design evaluates whether reasoning models can strategically select parameters to generate problems at specified difficulty levels.

**Spatial Reasoning.** Figure 7 illustrates an example of a sample from the spatial reasoning environment. On such samples, we ask 4 types of queries. i) Absolute location (x,y) co-ordinates of the particle or the board. The board's location is defined as the location of its center. ii) The tile number on which a specific particle is located. iii) The orientation of a given particle (north, east, west, or south), and iv) the relative location of a particle or board with respect to another particle or board. When an LLM is prompted with such problems, we instruct it to produce structured outputs along with its reasoning traces. The structured output is verified easily with the ground truth computed programmatically.

The parameter space includes `board_size`, an integer between 5 and 100. Boolean flags `board_rotates`, `particle_rotates`, `board_moves`, and `particle_moves` indicating whether board and particle rotations and movements are allowed or not. If particle rotations are allowed, then `allowed_particle_rotations` should be a non-empty subset of $\{0, 90, 180, 270, 360\}$, where each of these numbers indicates counter-clockwise rotation in degrees. If particle movements are allowed, then `allowed_particle_movements` should be a non-empty subset of $\{$LEFT, RIGHT, FORWARD, BACKWARD$\}$, indicating the entity moves 1 unit in the stipulated direction w.r.t its orientation (see Figure 7). Similarly, `allowed_board_rotations`

and `allowed_board_movements` should be set if their corresponding flags are on; otherwise, they should be empty sets. The parameter space also includes the numbers of each kind of actions to be applied, i.e., `number_of_board_rotations`, `number_of_particle_rotations`, `number_of_board_movements`, and `number_of_particle_movements`. Each of these must range between 0 to 15. Lastly, a flag `wrap_around` indicates whether the board's boundaries allow the overflowing movement of a particle to wrap around from the opposite side.

The descriptions of parameters and actions are provided in the prompt (Appendix B) for the designer model.

**Human Designed $\tau$-bench Airline.** The parameter descriptions and expected behaviors are specified in the designer prompt (Appendix B). Each sample corresponds to an airline itinerary planning scenario parameterized by a small set of discrete controls. The parameter space includes numerical factors such as `num_actions` (1–6), `num_passengers` (1–3), and `num_baggages` (0–3), as well as categorical attributes like `booking_strategy` ("cheapest"/"earliest_arrival"), `is_direct`, `is_round_trip`, `cabin` ("economy"/"business"), and `insurance` ("yes"/"no"). These parameters jointly control itinerary complexity: increasing action count, passengers, or bags expands the combinatorial search space, while enabling multiple strategies, connecting flights, or round-trip requirements adds additional reasoning constraints. When prompted with such parameterized tasks, the LLM designer is instructed to output both a *thought process* describing how does the configuration achieve the target failure rate and the final parameter values in structured JSON. This structured output can be programmatically validated against the student model's measured failure rate.

### Opus 4.1 Designed $\tau$-bench Airline.

The parameter space in the Opus 4.1 designed $\tau$-Bench extends beyond structural complexity (e.g., `num_actions` $\in [1, 6]$, `num_passengers` $\in [1, 3]$) to include behavioral and informational dimensions. Categorical controls specify booking preferences (`booking_strategy`: "cheapest"/"earliest_arrival"), routing options (`is_direct`, `is_round_trip`), cabin composition (`cabin_mix`: economy, business, or mixed), and environment conditions such as `information_completeness` (whether all data is provided upfront), `information_pattern` (upfront, gradual, reactive revelation of details), `cooperation_level` (helpful/demanding/uncooperative agents), and `preference_clarity` (explicit vs. implicit preferences). Together, these parameters modulate combinatorial difficulty, reasoning burden, and dialogue complexity, allowing fine-grained control of task hardness to steer the target model's empirical failure rate towards the target. The designer model receives a target failure rate $\rho_{\text{fail}}$ and is asked to generate task parameters that achieve $1 - \text{pass@}1 \approx \rho_{\text{fail}}$. Structured outputs include both the parameter configuration and a *thought process* explaining why it should achieve the desired difficulty level.

## A.2 DETAILED BASELINES

**RS+PPR.** The parameter $p$ is the probability to sample from the buffer of *good* parameters and $\Delta$ is the gap below which the parameters are considered *good*. We use $p = 0.5$ and $\Delta = 0.1$ in all the settings.

**BoN-ML Model Training and Selection.** As part of the BoN-ML experiments, we trained and compared classical machine learning models to predict regret efficiently. Across all three domains, we explored over 800 different parameter configurations and architectures. Given the relatively small datasets (100 samples per domain), with feature counts ranging from 13 to 74, we applied 5-fold cross-validation to obtain reliable performance estimates.

All features were derived directly from the environment parameters, ensuring the predictors remained lightweight and domain-specific. Models were selected based on the highest cross-validation R² score, and the best candidates were saved for deployment. Performance was domain-specific: small neural networks performed best for Arithmetic Sequences, Random Forests excelled in Spatial Reasoning, and gradient boosting worked best for $\tau$-Bench. This process yielded fast, domain-tailored predictors to guide BoN-ML parameter selection effectively. Table 3 summarizes the cross-validation $R^2$ training results.

Table 3: BoN-ML regret prediction training results (5-fold CV on 100 samples per domain).

| Domain | Features | Best Model | CV R² |
|---|---|---|---|
| Arithmetic Seq. | 74 | Neural Network ($74 - 2 - 1$, $\alpha$=1.0) | $0.52 \pm 0.08$ |
| Spatial Reasoning | 28 | Random Forest ($n$=50, $d$=10) | $0.43 \pm 0.05$ |
| $\tau$-Bench | 13 | Gradient Boosting ($n$=20, $lr$=0.1, $d$=2) | $0.17 \pm 0.24$ |

### A.3 DETAILS OF LLM MODELS

**LLM Versions** GPT-5: undisclosed - the latest GPT-5 version as of Sep 25, 2025 Opus 4.1: claude-opus-4-1-20250805 Grok 4: grok-4-0709 o4-mini: o4-mini-2025-04-16 claude3.7: claude-3-7-sonnet-20250219 gemini-2.5-flash: gemini-2.5-flash

**LLM Inference Parameters** The default temperature for designer models is 0.5, and for target models is 0.0. However, claude-opus-4-1-20250805 and claude-3-7-sonnet-20250219 are only available with a temperature of 1. On the Arithmetic Sequence, which is an agentic task, the target model uses a time horizon of 16 steps.

The default reasoning budget for designer models is 4096 tokens, and for target models is 1024. However, grok-4-0709 does not support a configurable reasoning budget.

### A.4 DATASET SIZES

During parameter search, the rollout dataset sizes are 10, 30, and 250 for Arithmetic Sequence, $\tau$-Bench, and Spatial Reasoning, respectively. For evaluation, we generate datasets using the selected parameters, with sizes of 75, 50, and 500 for Arithmetic Sequence, $\tau$-Bench, and Spatial Reasoning, respectively.

### A.5 CONVERGENCE RATE ANALYSIS

Figure 8 shows the rolling standard deviation (3-iteration window) of performance gap across all teacher models and target difficulty levels, comparing BeTaL and RS+PPR convergence stability. Despite vast differences in parameter spaces of the datasets, BeTaL consistently exhibits lower variability than RS+PPR. BeTaL 's standard deviation remains between 5-20% throughout parameter search, while RS+PPR maintains approximately 25% variability with minimal improvement over time. This demonstrates that BeTaL not only converges faster but does so more predictably and reliably.

At iteration $t$ (for $t \geq 2$), the rolling standard deviation uses a 3-iteration backward-looking window:

$$\sigma_{\text{roll}}(t) = \text{std}(G_{t-2} \cup G_{t-1} \cup G_t) \tag{1}$$

where $G_i$ represents performance gaps at iteration $i$ across all target accuracies, averaged across seeds within each teacher model. Results shown for $t \geq 2$ to ensure full windows.

### A.6 ADDITIONAL RESULTS

Figure 9 shows the observed average performance gaps when Claude Opus-4.1 and Grok-4 models are used as designer models. These results show BeTaL achieves low performance gaps across different designer models. We also provide a comparison of BeTaL with all designer models on all datasets and target difficulty levels in Figure 12. We see all three designer models achieve similar results across the settings.

We study the convergence behavior of iterative methods in settings ranging from trivial to hard difficulty levels (Figure 11). Except for a few settings, we see BeTaL iteratively improves its parameter estimates and converges to the desired performance gap after a few iterations. These results provide further evidence in support of LLMs' effectiveness as optimizers (Yang et al., 2024).

Next, in Figure 10 we show results over multiple evaluation models across different datasets. As the benchmarks were designed with o4-mini as the target model, we see a low performance gap when

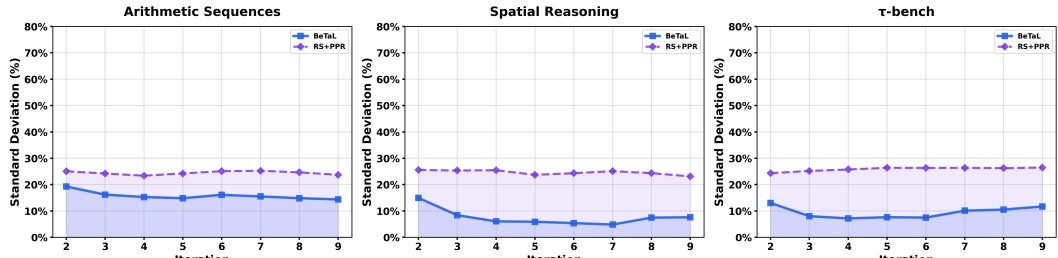

Figure 8: Three-iteration backward-looking rolling standard deviation comparing `BeTaL` and RS+PPR convergence stability across datasets (iterations 2-9), combined across all target accuracy levels. `BeTaL` (blue) exhibits consistently lower variability compared to RS+PPR (purple), indicating more stable and predictable convergence behavior.

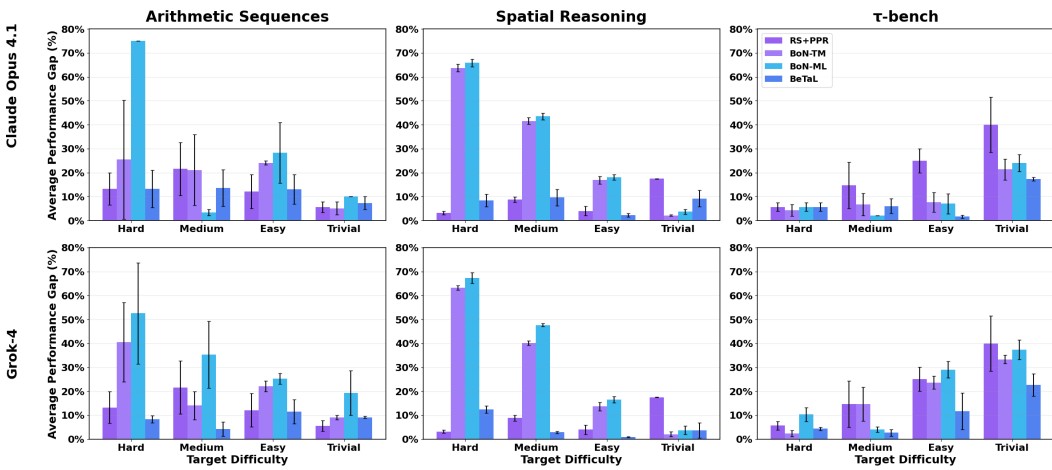

Figure 9: Evaluation results on o4-mini with `BeTaL` (with Claude Opus 4.1 or Grok-4 as the designer model, and o4-mini as the target model during parameter search) perform robustly at different target difficulty levels, compared to baselines on Arithmetic Sequences, Spatial Reasoning, and $\tau$-Bench.

evaluated on o4-mini. In the $\tau-$Bench setting, we see similar performance across different evaluation models. In the Spatial reasoning and Arithmetic sequences setups, there is a larger performance gap on evaluation models different from o4-mini; however, the range of observed accuracies (or regret) still reflects the relative hardness levels inherent in the benchmarks.

We also analyze the evolution of different parameters over the `BeTaL` iterations. Figures 13, 14, 15, 16 show the parameter evolution in the spatial reasoning setting with hard, medium, easy, and trivial difficulty levels, respectively. The results show the designer models start off with random (generally high) values of the parameters and gradually tweak them so that the performance gap is minimized. The evolution patterns for individual parameters matches with our intuitive understanding of the spatial reasoning environment. The models prefer larger board sizes, larger numbers and types of actions to increase the difficulty, and conversely smaller values to reduce the complexity. They also prioritize reducing/disabling board actions to reduce complexity, since an action on a board also triggers actions on the particles.

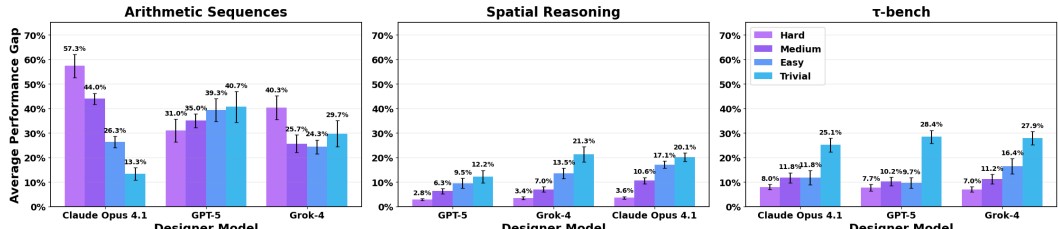

Figure 10: BeTaL performance by the designer model during parameter search across three benchmark domains. Each panel represents one dataset (Arithmetic Sequence, Spatial Reasoning, $\tau$-Bench) and compares three designer models (GPT-5, Grok-4, Opus-4.1) across four target performance levels: Hard ($\rho^{\text{hard}} = 0.25$), Medium ($\rho^{\text{medium}} = 0.50$), Easy ($\rho^{\text{easy}} = 0.75$), and Trivial ($\rho^{\text{trivial}} = 0.90$), shown as grouped bars. Bars show mean performance gap (difference between target and observed target performance), with o4-mini as target model, averaged over training iterations. Error bars show standard error.

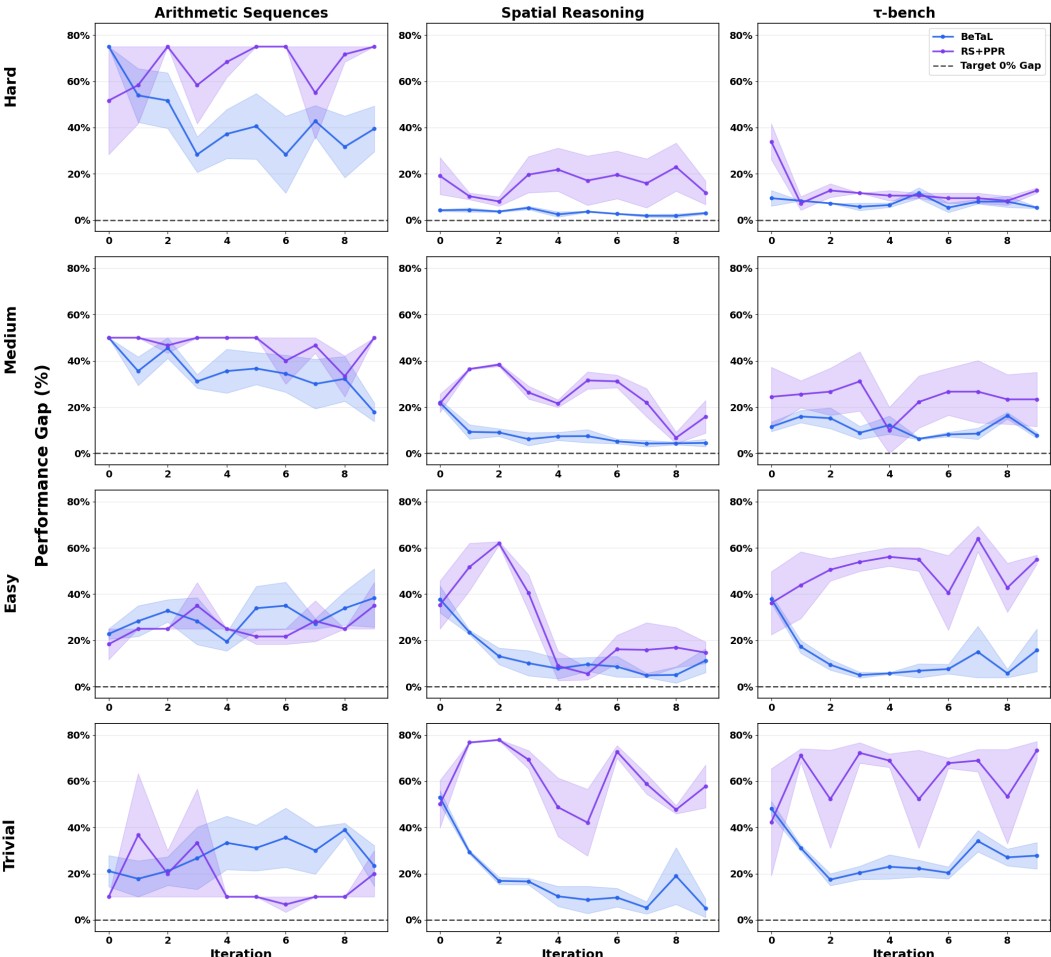

Figure 11: BeTaL vs. RS+PPR convergence during parameter search across datasets and target difficulty levels. Each panel shows the mean performance gap (difference between target and observed target performance) over training iterations for two design approaches: BeTaL (our method) and RS+PPR (baseline). Rows indicate target performance (Hard ($\rho^{\text{hard}} = 0.25$), Medium ($\rho^{\text{medium}} = 0.50$), Easy ($\rho^{\text{easy}} = 0.75$), and Trivial ($\rho^{\text{trivial}} = 0.90$)). Columns show three benchmark domains (Arithmetic Sequence, Spatial Reasoning, $\tau$-Bench). BeTaL results are averaged across designer models (GPT-5, Grok-4, Opus-4.1). All results use o4-mini as the target model, with shaded regions showing standard error across seeds.

Figure 12: Designer model performance during parameter search across datasets and target difficulty levels. Each panel shows the mean performance gap (difference between target and observed target performance) for different design approaches: BeTaL, BoN-ML, and BoN-TM. Rows indicate target performance (Hard ($\rho^{\text{hard}} = 0.25$), Medium ($\rho^{\text{medium}} = 0.50$), Easy ($\rho^{\text{easy}} = 0.75$), and Trivial ($\rho^{\text{trivial}} = 0.90$)). Columns show three benchmark domains (Arithmetic Sequence, Spatial Reasoning, $\tau$-Bench). All results use o4-mini as the target model, averaged over each iteration.

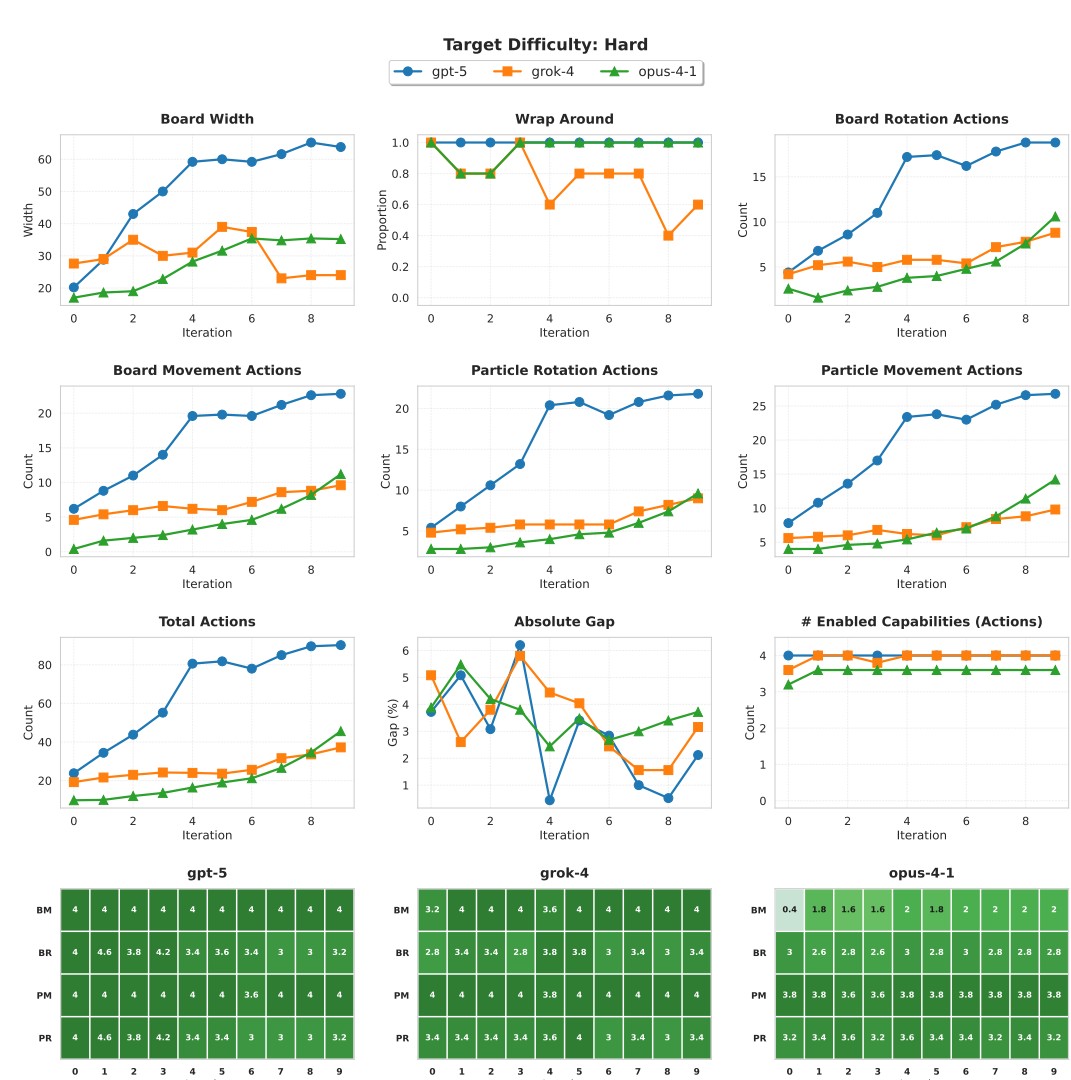

Figure 13: **Parameter evolution over iterations in the hard difficulty setting**. The subplots show average values of the different design parameters at each iteration chosen by the designer models (GPT-5, Grok-4, Opus-4-1). Row 1 shows board_size (width), wrap_around and number_of_board_rotation. Row 2 shows number_of_board_movements, number_of_particle_rotations and number_of_particle_movements. Next row presents the sum of these number of actions (total actions), absolute performance gap as observed on the o4-mini target model and the number of enabled capabilities (types of rotations and movements), here BM, BR are the sizes of sets allowed_board_movements and allowed_board_rotations and PM, PR similarly reflect the sizes of action sets corresponding to the particles. We can see to obtain a hard configuration, models generally prefer a larger board size and a higher number of capabilities and actions. Among the models, GPT-5 does it more aggressively and achieves the lowest performance gap as well.

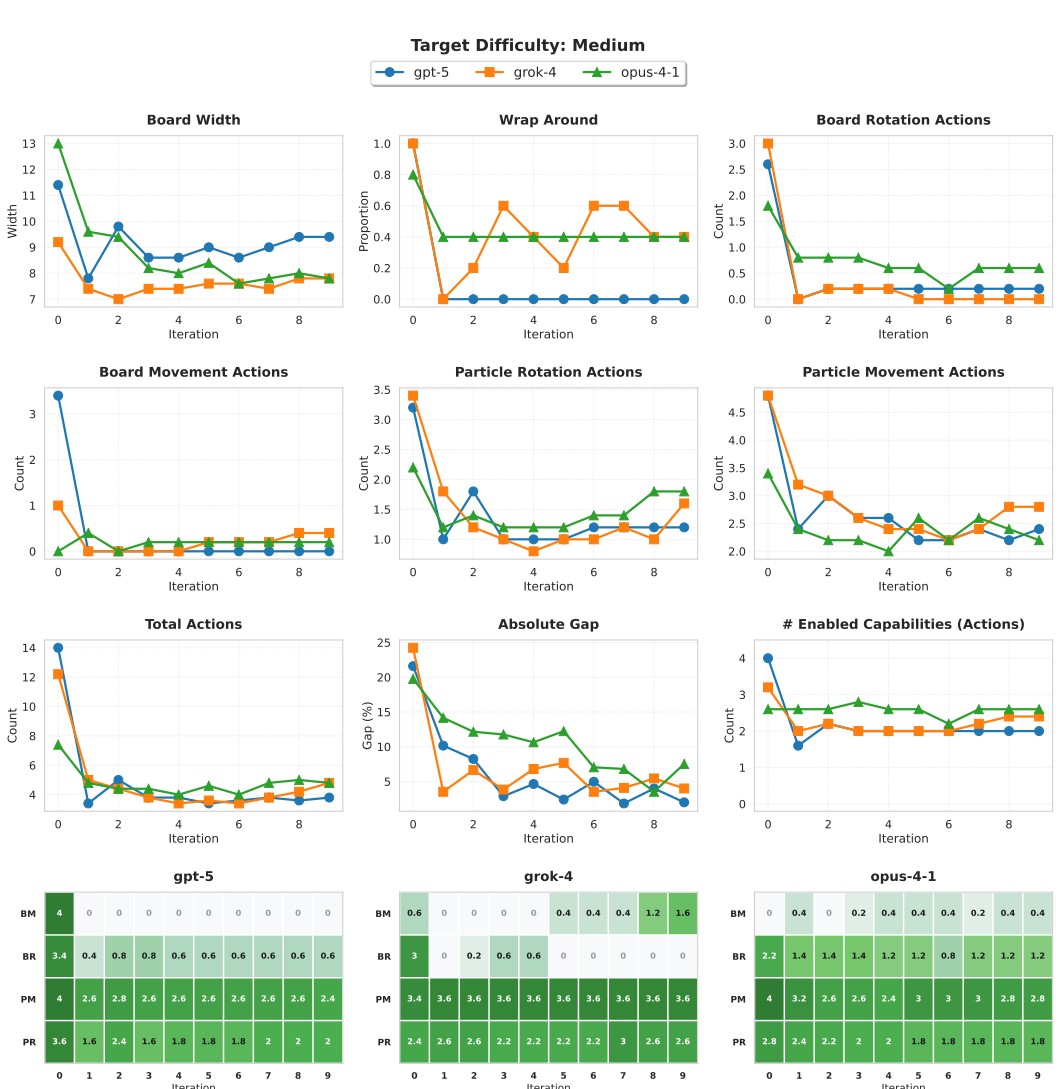

Figure 14: **Parameter evolution over iterations in the medium difficulty setting**. The sub-plots show average values of the different design parameters at each iteration chosen by the designer models (GPT-5, Grok-4, Opus-4-1). Row 1 shows board_size (width), wrap_around and number_of_board_rotation. Row 2 shows number_of_board_movements, number_of_particle_rotations and number_of_particle_movements. Next row presents the sum of these number of actions (total actions), absolute performance gap as observed on the o4-mini target model and the number of enabled capabilities (types of rotations and movements), here BM, BR are the sizes of sets allowed_board_movements and allowed_board_rotations and PM, PR similarly reflect the sizes of action sets corresponding to the particles. We can see that, to obtain a medium difficulty configuration, models prefer much smaller board sizes and number and types of actions as compared to the hard setting in Figure 13. Also consistent with the expectations, the models reduce the number of board actions close to 0 but allow a decent number of particle actions.

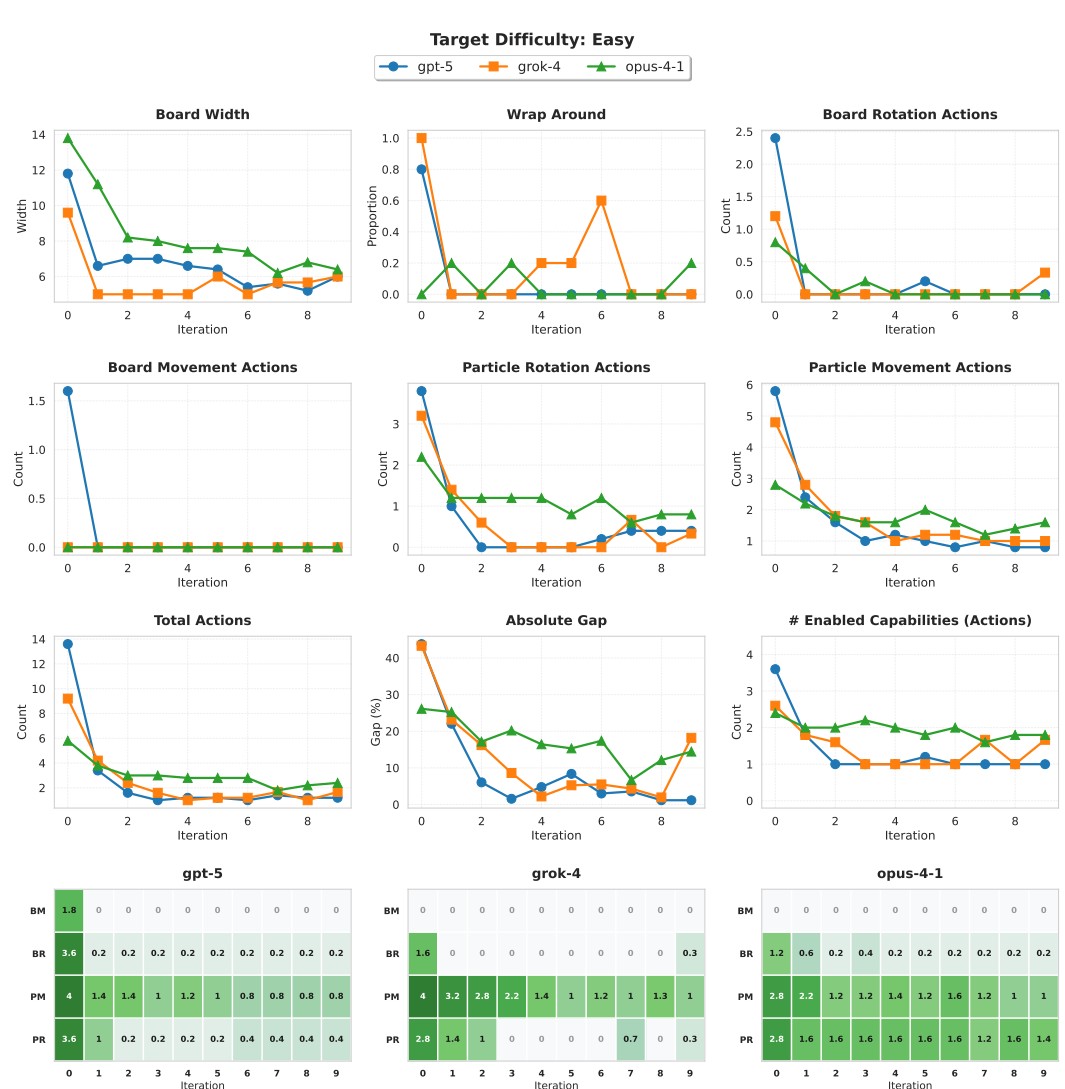

Figure 15: **Parameter evolution over iterations in the medium difficulty setting**. The subplots show average values of the different design parameters at each iteration chosen by the designer models (GPT-5, Grok-4, Opus-4-1). Row 1 shows board_size (width), wrap_around and number_of_board_rotation. Row 2 shows number_of_board_movements, number_of_particle_rotations and number_of_particle_movements. Next row presents the sum of these number of actions (total actions), absolute performance gap as observed on the o4-mini target model and the number of enabled capabilities (types of rotations and movements), here BM, BR are the sizes of sets allowed_board_movements and allowed_board_rotations and PM, PR similarly reflect the sizes of action sets corresponding to the particles. We can see, to obtain a medium difficulty configuration, models prefer much smaller board sizes and number and types of actions as compared to the hard setting in Figure 13. Also consistent with the expectations the models reduce the number of board actions close to 0 but allow a decent number of particles actions.

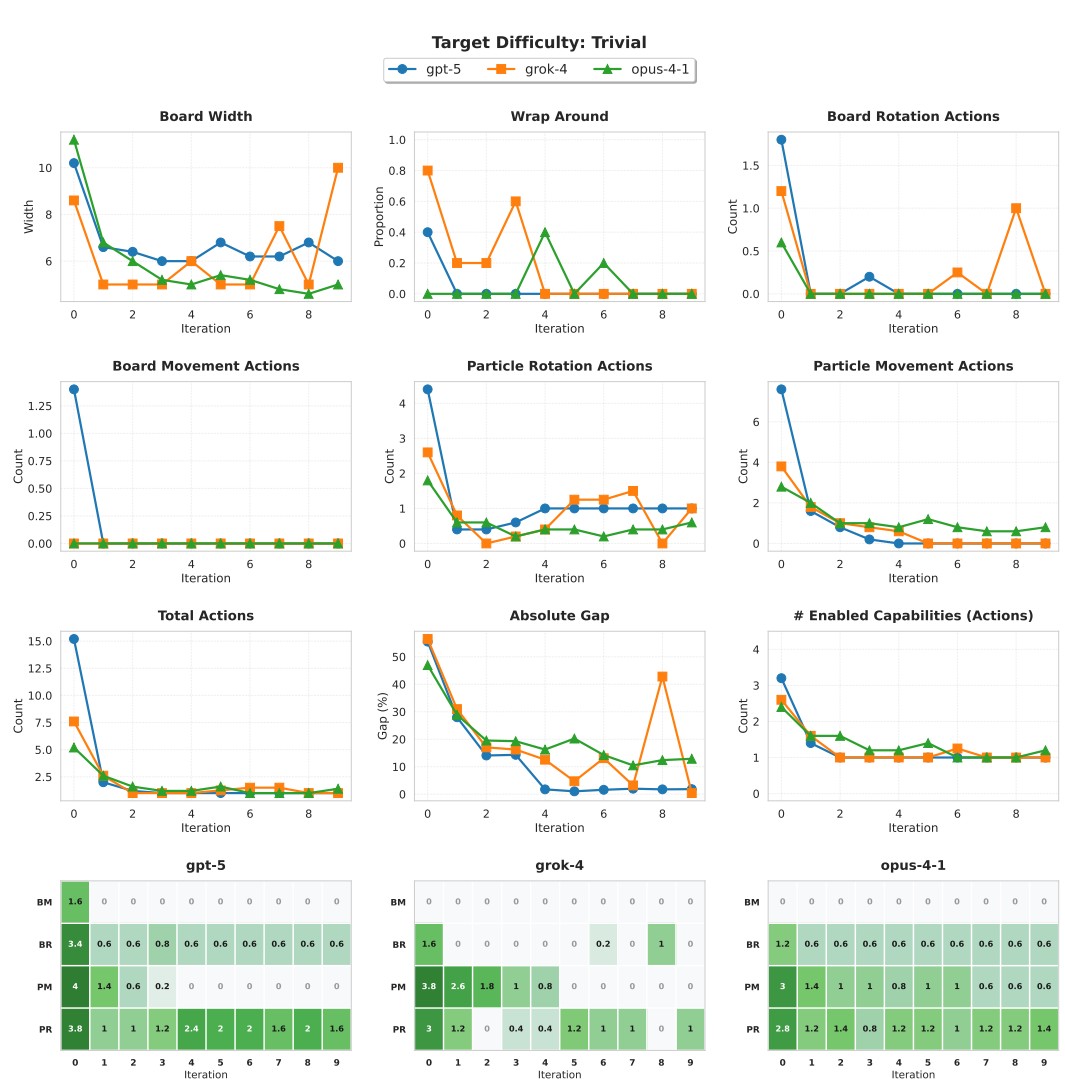

Figure 16: **Parameter evolution over iterations in the trivial difficulty setting**. The subplots show average values of the different design parameters at each iteration chosen by the designer models (GPT-5, Grok-4, Opus-4-1). Row 1 shows board_size (width), wrap_around and number_of_board_rotation. Row 2 shows number_of_board_movements, number_of_particle_rotations and number_of_particle_movements. Next row presents the sum of these number of actions (total actions), absolute performance gap as observed on the o4-mini target model and the number of enabled capabilities (types of rotations and movements), here BM, BR are the sizes of sets allowed_board_movements and allowed_board_rotations and PM, PR similarly reflect the sizes of action sets corresponding to the particles. We can see that, to obtain an easy difficulty configuration, models prefer smaller board sizes and number and types of actions as compared to the medium and easy settings in Figures 14 and 15. Also consistent with the expectations, the models reduce the number of board actions close to 0, but allow a few actions on particles.

# B PROMPTS

We provide the prompts provided to the designer models across the three tasks considered in the paper.

---

**LLM Designer Prompt for Arithmetic Sequence**

The math problem is to apply a sequence of operators on a number to produce a final answer. The sequence of operators are applied recursively on intermediate results, i.e., `num = operator(num)` for each operator in the sequence. The operators only take in one number as input.

You should target the given model regret at {`target_regret`}, so that the parameters can generate a math problem for the model at the desired regret level. A high regret indicates a challenging environment (1 for unsolvable), while a low regret indicates an easy environment (0 for easy).

Here is the feedback from the previous iterations, which you can use to generate new parameters: {`feedback`}

First, reason about the feedback from previous iterations. Specifically note what parameters/aspects made previous environments challenging or trivial.

Then, given a list of common math operators {`operators`}, your task is to generate values for the given parameters:

1. `feedback_summary`: your summary of the feedback from the previous iterations.

2. `thought_process`: your thought process for generating the parameters.

3. `max_range_of_nums`: the upper bound of range the input number can take on, i.e. (1, `max_range_of_nums`). Pick a number between 5 and 50.

4. `N`: the length of the sequence of operators to apply on a number (between 5 and 10)

5. `K`: The maximum number of times an operator can be repeated in the sequence (between 1 and 5)

6. `type_of_nums`: the type of numbers in the input (`int` or `float`)

7. `operator_sequence`: select 3 operators from the list above, to generate a sequence of operators of length `N` to apply on a number, where each operator can be repeated at most `K` times.

**Output format (JSON):**

```
{
    "feedback_summary": str,
    "thought_process": str,
    "max_range_of_nums": int,
    "N": int,
    "K": int,
    "type_of_nums": str,
    "operator_sequence": list[str]
}
```

---

**LLM Designer Prompt for the Spatial Reasoning Environment**

You are an expert in designing spatial reasoning environments. The environment is a 2D grid world. It consists of a square board and a two particles on the board. The board's dimensions can be from 5 to 100. The board is divided into tiles of size 1x1. The particles are at the center of the tiles.

Each object (board and particles) in the environment has an orientation and a location. The orientation is the direction in which the object is facing, which can be one of the following: NORTH, EAST, SOUTH, WEST. The location of particle is given by the 2D coordinates of the center of the tile on which the particle is located. The orientation and location of particle are initialized randomly. The location of the board is the 2D coordinates of the center of the board. The orientation of the board is the orientation of its center. It is always initialized to NORTH.

The environments complexity can be controlled by the following parameters:

- The board size determined by the width parameter.
- The board can either allow particles to wrap around the edges or not. It is determined by the wrap_around parameter. If it is true, then the particles can wrap around the edges of the board. If it is false, then the particles cannot wrap around the edges of the board.
- The movements allowed for the objects (board and particles). Each object can have a subset of the following movements: LEFT, RIGHT, FORWARD, BACKWARD.
- The rotations allowed for the objects (board and particles). Each object can have a subset of the following rotations: 0, 90, 180, 270, 360. If the rotation is 0, then the object is not rotated. If the rotation is 90, then the object is rotated 90 degrees counter-clockwise. If the rotation is 180, then the object is rotated 180 degrees counter-clockwise. If the rotation is 270, then the object is rotated 270 degrees counter-clockwise. If the rotation is 360, then the object is rotated 360 degrees counter-clockwise.

You are given a list of parameters for a board and a list of parameters for a particle. You are also given a list of parameters for actions that can be performed on the board and the particle. You need to design a spatial reasoning environment that is sufficiently challenging and an average language model can achieve a target accuracy of <accuracy>.

# Response format - JSON schema You must get the final answer and convert it to the following JSON data structure. Follow the schema exactly.

Key: thought_process
Type: String,
Description: Your thought process when designing the environment.

Key: 'wrap_around'
Type: Boolean,
Description: Whether the board can wrap around the edges.

Key: 'width'
Type: Integer,
Description: The width of the board.

Key: 'board_moves'
Type: Boolean,
Description: Whether the board can move.

Key: 'board_allowed_moves'
Type: List of Strings,
Description: The movements allowed for the board, must be subset of: LEFT, RIGHT, FORWARD, BACKWARD.

Key: 'board_rotates'
Type: Boolean,
Description: Whether the board can rotate.

Key: 'board_allowed_rotations'
Type: List of Integers,
Description: The rotations allowed for the board, must be subset of: 0, 90, 180, 270, 360.

Key: 'particle_moves'
Type: Boolean,

Description: Whether the particle can move.

Key: 'particle_allowed_moves'
Type: List of Strings,
Description: The movements allowed for the particle, must be subset of: LEFT, RIGHT, FORWARD, BACKWARD.

Key: 'particle_rotates'
Type: Boolean,
Description: Whether the particle can rotate.

Key: 'particle_allowed_rotations'
Type: List of Integers,
Description: The rotations allowed for the particle, must be subset of: 0, 90, 180, 270, 360.

Key: 'number_of_board_rotation_actions'
Type: Integer,
Description: The number of times the board can be rotated if board_rotates is true.

Key: 'number_of_particle_rotation_actions'
Type: Integer,
Description: The number of times the particles can be rotated if particle_rotates is true.

Key: 'number_of_board_movement_actions'
Type: Integer,
Description: The number of times the board can be moved if board_moves is true.

Key: 'number_of_particle_movement_actions'
Type: Integer,
Description: The number of times the particles can be moved if particle_moves is true.

---

**LLM Designer Prompt for $\tau$-bench Airline Environment**

You are an expert in designing airline-booking tasks for language-model agents.

Your goal is to propose task parameters that drive a student model to a target failure rate of 0.75. Here, the failure rate is defined as $1-\text{pass@1}$ for the student model, i.e., the probability that the student fails to solve the task on the first attempt. You are directly rewarded for minimizing the absolute distance to the 0.75 failure rate, so choose parameters that make the task challenging enough to approach this target as closely as possible.

The task setting is an airline-shopping environment where an agent must construct an itinerary subject to constraints (e.g., number of actions, bags, cabin class, routing).

Controllable parameters and intended complexity effects:

- `num_actions (1--6)`: Upper bound on primitive planning/interaction steps. Start simple with fewer actions; increase to raise difficulty.

- `num_passengers (1--3)`: More passengers typically increases combinatorial constraints and price/timing trade-offs.

- `num_baggages (0--3)`: More bags interact with fare rules and cabin choices; higher values generally increase difficulty.

- `booking_strategy`: Subset of {"cheapest", "earliest_arrival"}. Multiple strategies introduce objective trade-offs.

- `is_direct`: Boolean. Allowing `false` admits connections and routing search complexity.

- `is_round_trip`: Boolean. Round-trips add coupling between outbound/return constraints.

- cabin: Subset of {"economy", "business"}. More options broaden fare/rule search space.
- insurance: One of {"yes", "no"}. Insurance interacts with cost-focused strategies and can add goal ambiguity.

Tune these parameters to steer the student model's $1 - \text{pass@}1$ toward $0.75$.

# Response format — JSON schema
You must get the final answer and convert it to the following JSON data structure. Follow the schema exactly.

Key: thought_process
Type: String
Description: Concise reasoning explaining how the chosen parameters are expected to yield a failure rate near $0.75$; reference how each parameter affects difficulty.

Key: num_actions
Type: Integer (range: 1–6)
Description: Maximum number of allowed actions/steps.

Key: num_passengers
Type: Integer (range: 1–3)
Description: Number of travelers to book.

Key: num_baggages
Type: Integer (range: 0–3)
Description: Total checked bags across passengers.

Key: booking_strategy
Type: List of Strings (subset of: {"cheapest", "earliest_arrival"})
Description: Allowed objective(s) for the student; may include one or both.

Key: is_direct
Type: Boolean
Description: If true, only nonstop itineraries are valid; if false, connections are allowed.

Key: is_round_trip
Type: Boolean
Description: Whether the itinerary must include return travel.

Key: cabin
Type: List of Strings (subset of: {"economy", "business"})
Description: Allowed cabin classes.

Key: insurance
Type: String (one of: "yes", "no")
Description: Whether trip insurance is part of the task constraints.

---

**LLM Designer Prompt for $\tau$-bench Airline Environment (Parameter Space Generated by Opus-4.1)**

You are an expert in designing airline-booking tasks for language-model agents.

Your goal is to propose task parameters that drive a student model to a **target failure rate of {target_failure_rate}**. Here, the failure rate is defined as $1 - \text{pass@}1$ for the student model, i.e., the probability that the student fails to solve the task on the first attempt. You are directly rewarded for minimizing the absolute distance to {target_failure_rate}, so choose parameters that make the task challenging enough to approach this target as closely as possible.

The task setting is an airline-shopping environment where an agent must construct an itinerary subject to constraints (e.g., number of actions, passengers, bags, cabin class, routing, information flow, and user cooperation).

Controllable parameters and intended complexity effects:

- `num_actions (1--6):` Upper bound on primitive planning/interaction steps. Fewer actions constrain search; increasing actions raises planning depth and error surface.

- `num_passengers (1--3):` More travelers increase combinatorial constraints (seat availability, fares), amplifying trade-offs.

- `num_baggages (0--3):` More bags interact with fare rules and cabin choices; higher values generally increase difficulty.

- `booking_strategy:` Subset of {"cheapest", "earliest_arrival"}. Multiple objectives introduce competing trade-offs and ambiguity.

- `is_direct:` Boolean. Allowing `false` admits connections and routing search complexity (layovers, MCT).

- `is_round_trip:` Boolean. Round-trips couple outbound/return constraints and calendaring.

- `cabin_mix:` Subset of {"economy_only", "business_only", "mixed"}. `mixed` broadens fare/rule search and cross-cabin reasoning.

- `information_completeness:` Boolean. If `false`, key facts are omitted initially, forcing clarification steps and robustness to uncertainty.

- `cooperation_level:` Subset of {"helpful", "demanding", "uncooperative"}. Less cooperative users increase dialogue turns, constraint changes, and error likelihood.

- `information_pattern:` Subset of {"upfront", "gradual", "reactive"}. Non-upfront patterns stagger constraints and increase planning revisions.

- `preference_clarity:` Subset of {"explicit", "implicit"}. `implicit` requires inference from hints (e.g., times, budgets), increasing ambiguity.

Tune these parameters to steer the student model's $1 - \text{pass}@1$ toward {`target_failure_rate`}.

# Response format — JSON schema
You must get the final answer and convert it to the following JSON data structure. Follow the schema exactly.

Key: `thought_process`
Type: String
Description: Concise reasoning explaining how the chosen parameters are expected to yield a failure rate near {`target_failure_rate`}; reference how each parameter affects difficulty.

Key: `num_actions`
Type: Integer (range: 1–6)
Description: Maximum number of allowed actions/steps.

Key: `num_passengers`
Type: Integer (range: 1–3)
Description: Number of travelers to book.

Key: `num_baggages`
Type: Integer (range: 0–3)
Description: Total checked bags across passengers.

Key: `booking_strategy`
Type: List of Strings (subset of: {"cheapest", "earliest_arrival"})
Description: Allowed objective(s) for the student; may include one or both.

Key: `is_direct`
Type: Boolean
Description: If `true`, only nonstop itineraries are valid; if `false`, connections are allowed.

Key: `is_round_trip`
Type: Boolean
Description: Whether the itinerary must include return travel.

Key: `cabin_mix`
Type: List of Strings (subset of: {"economy_only", "business_only", "mixed"})
Description: Allowed cabin configuration scope.

Key: `information_completeness`
Type: Boolean
Description: If `true`, all necessary details are provided initially; if `false`, some are withheld.

Key: `cooperation_level`
Type: List of Strings (subset of: {"helpful", "demanding", "uncooperative"})
Description: Expected user cooperation profile(s).

Key: `information_pattern`
Type: List of Strings (subset of: {"upfront", "gradual", "reactive"})
Description: How and when information is revealed during the interaction.

Key: `preference_clarity`
Type: List of Strings (subset of: {"explicit", "implicit"})
Description: Whether preferences are stated clearly or must be inferred.

---

## Example of a Question on the Arithmetic Sequence Task

You are an agent that can use tools via tool calling:
If you have the final answer, respond with: FINAL ¡sequence of operators as a comma separated list¿

Given the following input number and final answer, use the functions provided to perform the correct sequence of operations on the input number to get the final answer.

Input number: 2.4460677252452125

Final answer: 4.423634456186643

---

## Example of a Question in the Spatial Reasoning Setting

Following is the description of the spatial reasoning environment. Go through it carefully and then answer the question in the requested format.

# Environment

## Setup
All locations are pairs of real numbers (x, y). North corresponds to increasing y, and South corresponds to decreasing y. East corresponds to increasing x, and West corresponds to decreasing x. Orientation is a direction, and can be one of the following: North, East, South, or West. Orientation is also measured in degrees, and can be one of the following: 0, 90, 180, 270. Where 0 means East, 90 means North, 180 means West, and 270 means South.

A board's rotation is defined as the rotation of the board around its center. When a board rotates, the orientation of the board changes, and the tiles and particles on the board also rotate along with it. A particle's rotation changes the orientation of the particle, but does not

change the location of the particle. As a general rule, any entity's rotation can change the orientation of the entity, but does not change the location of the entity.

A board's location is defined as the location of its center. A board's movement changes the location of the board, and the tiles and particles on the board also move along with it. For example, if a board moves forward 1 unit, the center of the board and the tiles and particles on the board all move 1 unit along the orientation of the board. A particle's movement changes the location of the particle For example, if a particle moves forward 1 unit, the location of the particle changes by 1 unit along the orientation of the particle.

If the movement of particles results in the particle moving beyond the boundary of the board, then the particle will either wrap around the boundary of the board or remain at the current tile. It depends on the board's wrap around settings, which are described in the description of the board. As a general rule, any entity's movement can change the location of the entity, but does not change the orientation of the entity. The orientation of an entity can be thought of as the direction in which the entity is facing. This determines the meaning of forward, backward, left, right, etc., for the entity.

## Entities

The environment contains the following entities:

# Board B1

## Setup A board is 12.0 units wide and 12.0 units tall, and contains 2 particle(s). It is centered at (0.0, 0.0). Its orientation is defined as the center's orientation, which is NORTH. Initially, the board is oriented NORTH.
The board has four sides: SIDE-1, SIDE-2, SIDE-3, SIDE-4 The side from the south west corner to south east corner is the bottom side of the board. It is called SIDE-1 The side from the south east corner to north east corner is the right side of the board. It is called SIDE-2 The side from the north east corner to north west corner is the top side of the board. It is called SIDE-3 The side from the north west corner to south west corner is the left side of the board. It is called SIDE-4

## Boundaries

In the event the particle move results in the particle moving beyond the boundary of the board, the resulting location is decided as follows:

When a particle is on a tile, it means its location is the tile's centroid. The SIDE-1 of the board can be crossed when approaching from the SIDE-3, and the particle(s) will move to the opposite tile on the SIDE-3. The SIDE-2 of the board can be crossed when approaching from the SIDE-4, and the particle(s) will move to the opposite tile on the SIDE-4. The SIDE-3 of the board can be crossed when approaching from the SIDE-1, and the particle(s) will move to the opposite tile on the SIDE-1. The SIDE-4 of the board can be crossed when approaching from the SIDE-2, and the particle(s) will move to the opposite tile on the SIDE-2.

## Tiles on the board

The board is divided into square tiles of size 1 units by 1 units. Tiles are numbered from 1 to (width * height), starting from the bottom left corner in a zigzag pattern. Going from left to right, then right to left, and so on. For example, for a 3x3 board, the tiles are numbered as follows: 9 8 7 6 5 4 1 2 3

## Allowed moves

The following moves are allowed for the board: FORWARD - board moves forward 1 unit. BACKWARD - board moves backwards 1 unit. Orientation remains the same. LEFT - board sidesteps 1 unit to the left. Orientation remains the same. RIGHT - board sidesteps 1 unit to the right. Orientation remains the same.

## Allowed rotations

The following rotations are allowed for the board: 90 - board rotates 90 degrees. 180 - board rotates 180 degrees. 270 - board rotates 270 degrees.

# Particle P1

## Initial State

It is located at (3.5, 3.5), and is facing WEST (180 degrees). It is on tile 111. It is on board B1.

## Allowed moves

The following moves are allowed for this particle: FORWARD - particle moves forward 1 unit. BACKWARD - particle moves backwards 1 unit. Orientation remains the same. LEFT - particle sidesteps 1 unit to the left. Orientation remains the same. RIGHT - particle sidesteps 1 unit to the right. Orientation remains the same.

## Allowed rotations

The following rotations are allowed for this particle: 90 - particle rotates 90 degrees. 180 - particle rotates 180 degrees. 270 - particle rotates 270 degrees.

# Particle P2

## Initial State

It is located at (-0.5, 5.5), and is facing SOUTH (270 degrees). It is on tile 139. It is on board B1.

## Allowed moves

The following moves are allowed for this particle: FORWARD - particle moves forward 1 unit. BACKWARD - particle moves backwards 1 unit. Orientation remains the same. LEFT - particle sidesteps 1 unit to the left. Orientation remains the same. RIGHT - particle sidesteps 1 unit to the right. Orientation remains the same.

## Allowed rotations

The following rotations are allowed for this particle: 90 - particle rotates 90 degrees. 180 - particle rotates 180 degrees. 270 - particle rotates 270 degrees.

# Actions

The actions are the following: First, board B1 is rotated by 270 degrees. Then, particle P2 is rotated by 270 degrees. Then, particle P1 is rotated by 270 degrees. Then, particle P1 is rotated by 90 degrees. Finally, move particle P2 BACKWARD by 1 units.

# Question

What is the location of board B1 after all the actions?

# Response format - JSON schema You must get the final answer and convert it to the following JSON data structure. Follow the schema exactly.

Key: 'board_B1_x'
Type: Float,
Description: The x-coordinate of board B1 after all the actions.

Key: 'board_B1_y'
Type: Float,
Description: The y-coordinate of board B1 after all the actions.

---

**Example of a Task in the $\tau$-bench Airline Setting**

Following is the description of the airline environment. Go through it carefully and then answer the question in the requested format.

# Environment

## Setup
The environment simulates a commercial airline booking system. Airports are identified by IATA codes (e.g., SEA, EWR). Dates are formatted `YYYY-MM-DD`. Times are `HH:MM:SS` in local (EST) for scheduling metadata. Cabins include `basic_economy`, `economy`, and `business`. Bookings may be `one_way` or `round_trip`. Payment instruments include `certificate`, `gift_card`, and `credit_card`. Baggage may be free or non-free depending on fare rules (not shown here). Insurance is optional.

## Capabilities
Agents may:

- Search flights (nonstop or onestop) between an origin and a destination on a specified date.
- Book reservations with specified flight legs, cabin, passengers, baggages, insurance, and payment methods (in priority order).
- Request issuance of a travel certificate with a specified ID and amount.

# Entities

## User U1
User identifier: `mohamed_li_7869`.
The user's birthday is present in the profile and should not be requested during the interaction.

## Passenger(s)
A single passenger is provided and known to the user:

- `first_name`: Yusuf,  `last_name`: Thomas,  dob: 1966-05-11

## Payment Instruments (available to U1)

- `gift_card_3525913`: amount 27
- `gift_card_5876000`: amount 176
- `gift_card_7716568`: amount 237
- `credit_card_1922786`: amount 139

Preferred payment order: certificate → gift card → credit card.

# Demands

## Demand 1: Flight Search
Search for an onestop flight from `LGA` to `DTW` on `2024-05-25`.

## Demand 2: Booking
Book a one-stop, one-way itinerary from `SEA` to `EWR` on `2024-05-30` in business cabin for 1 passenger with 1 total baggage. Choose the cheapest eligible option. Include insurance. Use payments in the order: certificate(s) first, then gift card(s), then credit card(s).
Candidate flights presented (for selection during booking):

- Leg 1:
    - `flight_number`: HAT117,  origin: SEA,  destination: DFW
    - `scheduled_departure_time_est`:                    10:00:00, `scheduled_arrival_time_est`: 14:00:00
    - `status`: available,  date: 2024-05-30
    - Seats available: basic_economy 5,  economy 0,  business 1
    - Prices: basic_economy 62,  economy 119,  business 263
- Leg 2:
    - `flight_number`: HAT063,  origin: DFW,  destination: EWR

– `scheduled_departure_time_est:` 18:00:00, `scheduled_arrival_time_est:` 21:30:00
– `status:` available, `date:` 2024-05-30
– Seats available: basic_economy 11, economy 15, business 9
– Prices: basic_economy 80, economy 137, business 286

## Demand 3: Certificate Issuance
Request a certificate with:

- `certificate_id: certificate_4314319`
- `amount: 170`

# Actions
The intended agent actions, in order, are as follows:

1. search_onestop_flight with {`origin:` PHX, `destination:` DFW, `date:` 2024-05-18}.

2. book_reservation with the provided passenger, baggage, cabin, flight legs (SEA→DFW, then DFW→EWR on 2024-05-30), one-way, business, cheapest, insurance yes, and payment methods listed above in the stated priority order.

3. send_certificate with {`certificate_id: certificate_4314319`, `amount:` 170}.

*Note.* Although Demand 1 specifies LGA→DTW (2024-05-25) search, the sample action shows PHX→DFW (2024-05-18). The agent must honor the stated Demands when resolving inconsistencies (prefer Demands).

# Question
Produce the exact JSON payload(s) for the three API calls in the correct order that satisfy all Demands above (use LGA→DTW for the search as specified by Demand 1; for booking, choose the cheapest eligible `business` one-stop SEA→EWR itinerary from the two legs provided; include insurance; and apply payment instruments in the order certificate → gift card(s) → credit card(s)).

# Response format - JSON schema
Return a single JSON object with the following keys:

Key: `action_sequence`
Type: Array of Objects
Description: The ordered list of actions. Each object must have:

- `name` (string; one of `search_onestop_flight`, `book_reservation`, `send_certificate`)
- `kwargs` (object; the exact arguments for the call)

Key: `notes`
Type: String
Description: Brief justification for flight choice, insurance inclusion, baggage count, and the payment breakdown order.

## C  TEACHER MODEL REASONING TRACES

This appendix presents reasoning traces from teacher models designing environments across multiple tasks. Each example shows the Chain-of-Thought reasoning used by the teacher model to select environment parameters targeting specific difficulty levels.

### C.1  ARITHMETIC SEQUENCE

---

**Example 1: Grok-4**

**Method:** `BeTaL`
**Target Accuracy:** 25.0%
**Task:** Arithmetic Sequence

*Teacher Model Reasoning at the 3rd iteration:*

> **Feedback Summary:** The previous iterations had accuracy of 100.0%, indicating trivial environments. The sequences involved repeating patterns of pow, mul, and sqrt or div, which led to cyclic or stable values that were easy to compute.
>
> **Thought Process:** The previous environments used float numbers with operator sequences that resulted in repetitive and predictable patterns, leading to accuracy of 100.0%. The computations were simple due to cycling between values. To target a accuracy of 25%, I will use integer numbers and a sequence consisting of many squaring operations using 'pow' and 'mul' to produce extremely large numbers that are difficult for the model to compute exactly. I choose $N = 8$, $K = 4$, max_range_of_nums $= 40$, type $= 'int'$, operators 'pow', 'mul', 'sqrt', but the sequence will use only 'pow' and 'mul' to avoid reducing the number. The sequence will have 4 'pow' and 4 'mul' for 8 squarings, leading to $x^{256}$, which for $x \approx 40$ has about 411 digits, which should be challenging.

---

**Example 2: GPT-5**

**Method:** BoN-TM
**Target Accuracy:** 25.0%
**Task:** Arithmetic Sequence

*Teacher Model Reasoning:*

Goal: Set parameters so that deducing the operator sequence from a single input-output example is challenging but solvable (target accuracy $\sim 0.25$). I will iterate up to two times.

Iteration 1 (rejected): - Tried ops: {pow, div, sqrt}, $N = 8$, $K = 2$. Sampled a sequence mixing pow and sqrt. Observation: pow and sqrt largely cancel or explode values; multiple reverse paths produce identical results, leading to high ambiguity. Estimated accuracy $\sim 0.1$ (too hard).

Iteration 2 (accepted): - Choose ops: {mul, add, sqrt}. Set $N = 7$, $K = 3$ so each operator can appear at most 3 times. This mix gives nonlinearity via sqrt and moderate growth via mul/add, making reverse inference nontrivial but feasible. - For sampling only, assume unary meanings commonly used in such puzzles: add = $+1$, mul = $\times 2$, sqrt = principal square root. Type: float to accommodate non-perfect-square roots. Max input range cap: 20. - Constructed operator_sequence of length 7 with counts within K: [mul, add, sqrt, mul, add, sqrt, mul] (mul$\times 3$, add$\times 2$, sqrt$\times 2$). - Sample forward calculation (input = 9): 1) mul: $9 \to 18$ 2) add: $18 \to 19$ 3) sqrt: $19 \to \sim 4.3588989435$ 4) mul: $\sim 4.3589 \to \sim 8.7177978871$ 5) add: $\sim 8.7178 \to \sim 9.7177978871$ 6) sqrt: $\sim 9.7178 \to \sim 3.1189801054$ 7) mul: $\sim 3.1190 \to \sim 6.2379602108$ (final) - Reverse reasoning feasibility (given input 9, final $\sim 6.23796$, $N = 7$, ops $\in$ {add, mul, sqrt}): Assume last op mul $\to$ divide by 2: $\sim 3.11898$; prior sqrt $\to$ square: $\sim 9.7178$; prior add $\to$ subtract 1: $\sim 8.7178$; prior mul $\to$ divide by 2: $\sim 4.3589$; prior sqrt $\to$ square: 19; prior add $\to$ subtract 1: 18; prior mul $\to$ divide by 2: 9 = input. This reconstructs a consistent and plausible unique path under constraints, but a model would need nontrivial backtracking/search, making it challenging. - Difficulty assessment: Branching factor up to 3 with numeric constraints and float noise; sequence length 7; inverse search with sanity checks is required. This is difficult but solvable. Estimated accuracy $\approx 0.25$.

Conclusion: Use the parameters below with operator_sequence limited to {mul, add, sqrt}, $N = 7$, $K = 3$, float numbers, and input range up to 20 to achieve the target challenge level.

**Example 3: Claude Opus 4.1**

**Method:** BoN-ML
**Target Accuracy:** 25.0%
**Task:** Arithmetic Sequence

*Teacher Model Reasoning:*

Iteration 1: Starting with moderate complexity - Let's use $N = 6$, $K = 2$ with operators that create some ambiguity. - Testing: ['mul', 'sqrt', 'add'] with sequence ['mul', 'sqrt', 'add', 'mul', 'sqrt', 'add']. - With input 4:

$4 \rightarrow 8 \, (\text{mul} \times 2) \rightarrow 2.83 \, (\text{sqrt}) \rightarrow 3.83 \, (\text{add}+1) \rightarrow 7.66 \, (\text{mul} \times 2) \rightarrow 2.77 \, (\text{sqrt}) \rightarrow 3.77 \, (\text{add}+1)$

- This is somewhat traceable, maybe too easy for 0.25 accuracy.

Iteration 2: Increasing complexity - $N = 7$, $K = 3$, using operators that interact in complex ways. - Testing: ['pow', 'div', 'sqrt'] with sequence ['pow', 'sqrt', 'div', 'pow', 'sqrt', 'div', 'sqrt']. - With input 4:

$4 \rightarrow 16 \, (\text{pow}) \rightarrow 4 \, (\text{sqrt}) \rightarrow 2 \, (\text{div} \div 2) \rightarrow 4 \, (\text{pow}) \rightarrow 2 \, (\text{sqrt}) \rightarrow 1 \, (\text{div} \div 2) \rightarrow 1 \, (\text{sqrt})$

- The pow and sqrt operations create multiple possible paths, better for 0.25 accuracy.

Iteration 3: Fine-tuning for target accuracy - $N = 8$, $K = 3$, float type for added precision challenges. - Testing: ['mul', 'sqrt', 'pow'] — these operators create exponential growth/reduction patterns. - Sequence: ['mul', 'pow', 'sqrt', 'mul', 'sqrt', 'pow', 'mul', 'sqrt']. - This creates complex transformations that are difficult but not impossible to reverse-engineer. - The combination of squaring (pow) and square root operations creates ambiguity. - Multiple valid paths could potentially lead to the same result. - Estimated accuracy: $\sim 0.25$ (challenging but solvable with careful analysis).

## C.2 SPATIAL REASONING

**Example 4: Grok-4**

**Method:** BeTaL
**Target Accuracy:** 75.0%
**Task:** Spatial Reasoning

*Teacher Model Reasoning:*

To achieve a target accuracy of 75% for an average language model, I designed the environment with a 10x10 board and wrap-around enabled to introduce moderate complexity in position calculations. The board is allowed to rotate in 90 and 180 degrees up to 2 times, adding challenge to the reference frame without overwhelming. Particles can move in all four directions up to 5 times and rotate in 90, 180, 270 degrees up to 3 times each. This setup requires tracking multiple transformations, leading to potential errors in about 25% of cases.

---

**Example 5: Claude Opus-4.1**

**Method:** BoN-TM
**Target Accuracy:** 75.0%
**Task:** Spatial Reasoning

*Teacher Model Reasoning:*

> Designing for 75% accuracy (0.25 regret): Need moderate complexity with some challenging elements but not overwhelming. Key decisions: 12x12 grid provides good spatial complexity without being too large. Wrap-around adds edge-case complexity. Board rotation (90, 180) creates coordinate transformation challenges. Limited board movement (LEFT, RIGHT) adds another dimension without overwhelming. Particles have full movement but limited rotation for balanced complexity. Action counts moderate (2-3) to require multi-step reasoning but not excessive mental load.

---

**Example 6: Claude Opus-4.1**

**Method:** BoN-ML
**Target Accuracy:** 75.0%
**Task:** Spatial Reasoning

*Teacher Model Reasoning:*

> To achieve 75% accuracy (0.25 regret), I need moderate complexity. After iterating through options, I've selected an 8x8 grid which is large enough to be interesting but small enough to track mentally. The key challenge comes from board rotations (90, 180, 270 degrees) which transform the coordinate system, combined with particle movements (FORWARD/BACKWARD) relative to their orientation. With 2 board rotations and 3 particle movements, problems require tracking coordinate transformations and orientations through ~5 steps total, creating moderate difficulty suitable for the target performance level.

## C.3 $\tau$-BENCH

---

**Example 7: Opus-4.1**

**Method:** `BeTaL`
**Target Accuracy:** 75.0%
**Task:** $\tau$-bench

*Teacher Model Reasoning:*

> Based on the previous results, I observe that `num_actions=4` with 2 passengers and 2 baggages gave a failure rate of 0.7, while `num_actions=5` with 2 passengers and 3 baggages also gave 0.7. Both are 0.05 away from the target of 0.75. To potentially hit closer to 0.75, I'll try `num_actions=4` with 2 passengers but increase baggages to 3. This creates a configuration between the tested ones—slightly more complex than the 4-2-2 setup (which gave 0.7) but less complex than the 5-2-3 setup (which also gave 0.7). This intermediate complexity might push the failure rate slightly higher toward 0.75.

**Example 8: GPT5**

**Method:** BoN-TM
**Target Accuracy:** 75.0%
**Task:** $\tau$-bench

*Teacher Model Reasoning:*

> To target a 0.75 failure rate (pass@1 0.25), I increased task complexity via higher action count and maximum passengers, enabled multiple competing criteria and toggles (booking strategy, directness, round-trip, insurance), and limited one dimension (cabin) plus moderate baggage count to avoid overshooting difficulty.

**Example 9: Grok-4**

**Method:** BoN-ML
**Target Accuracy:** 75.0%
**Task:** $\tau$-bench

*Teacher Model Reasoning:*

> To target a failure rate of 0.75, I aim for moderate to high complexity. I set `num_actions` to 4 for some complexity without maxing out. `num_passengers` at 2 and `num_baggages` at 2 add moderate difficulty. Including both options in arrays for `booking_strategy`, `is_direct`, `is_round_trip`, `cabin`, and `insurance` creates varied and potentially challenging tasks, which could lead to around 75

**Note:** All reasoning traces show the teacher model's explanation of why specific parameters were chosen to achieve the target difficulty level. Different experiments may use "scratchpad" or "thought_process" field names due to prompt variations; both contain equivalent teacher model reasoning.

