# OpenReview forum: "Automating Benchmark Design"
_ICLR.cc/2026/Conference — Submitted to ICLR 2026_

### Official Review · Reviewer_gkRb · 2025-10-31

**Soundness:** 2
**Presentation:** 2
**Contribution:** 2
**Rating:** 4
**Confidence:** 3

**Summary:**

The paper presents a method for automated benchmark design with LLMs in the loop. The idea is that given a benchmark whose difficulty can be adjusted by changing some parameters, we can then automatically have LLMs find the right parameters so to achieve the right target difficulty, which is an important problem in the context of unsupervised environment design.

**Strengths:**

The idea to extend UED to automate full benchmark design with LLMs is interesting, and an important problem.

**Weaknesses:**

The environments being evolved are very toyish

The idea of UED from my understanding is to automatically create environments/tasks at the right level of difficulty for agents/policies to perform RL on. If the task is too easy or too difficult, the target agent/policy will not benefit from training in it. BeTaL attempts to solve the designer problem of creating tasks at the right difficulty, however it then does not attempt to have weaker models/policies train on those tasks, starting from the trivial ones, then increasing the difficulty as they improve. Given the narrative adopted by the paper and the analogies and comparisons with UED this seems like a missing experiment to me.

On many of the tasks the error bounds are very large, making it difficult to draw conclusions on performance differences. Running more seeds may help understanding actual performance differences. Since this is expensive, perhaps proving this in a single setup could be fine (eg: Run with #seeds >> 3 for a single model on all the benchmarks). Results on the t-bench are particularly weak.

I’m not particularly familiar with the space of automated benchmark design, but comparisons with bayesian optimization such as gaussian processes without LLMs in the loop might be a good simple baseline to try.

Editorials nits:
A few links for citations and figures are broken (lines 251, 399)

**Questions:**

1. Is it possible to run any curriculum learning experiments in real UED fashion, to check the validity of the method?
2. Is it possible to increase the number of seeds to improve the statistical significance of the results? Too high variance currently makes it a bit difficult to draw more conclusions.
3. How does the method compare to simpler bayesian optimization baselines without LLMs in the loop?

---

> ### Author Response · Authors · 2025-12-03
> **Replies to Reviewer gkRb**
>
> Thank you for your thoughtful and constructive feedback, and for recognizing the motivation and importance of our automated evaluation framework! We address your comments point by point below:
> > W1: Environment is toyish
>
> We respectfully disagree and clarify our environment choices.
>
> First, while Arithmetic Sequences is intentionally simple for controlled analysis, the other two environments are not toy: Spatial Reasoning is closely related to ARC-style pattern-induction benchmarks for compositional reasoning, and Tau-Bench Airline is built from **production, multi-step agent pipelines** involving realistic tool use and API interaction. This structured-but-realistic design is aligned with standard evaluation practice (e.g., SWE-Bench–style and GAIA–style settings).
>
> Second, these environments were chosen because they are parameterizable and scalable, with clear axes such as size of the problem space, number of reasoning steps, and action or tool complexity. These properties are shared by many real-world agentic and reasoning tasks and are meaningful axes along which to scale difficulty with.
>
> Finally, BeTaL is not limited to the specific environments we show: it only requires a parameterized task generator and a performance signal (execution, rules, or LLM-as-judge), making it directly applicable to richer coding, STEM, QA, and agent benchmarks beyond our current suite.
>
> > W2 & Q1: Training weaker models on in UED-style curriculum learning
>
> Thank you for identifying the connection between our framework, BeTaL, and curriculum learning approaches for RL. We share the excitement about this direction and agree that BeTaL may indeed extend beyond automated development of evaluation benchmarks, potentially informing curriculum learning strategies as well.
> While we also find this connection compelling, thoroughly exploring and validating curriculum learning for RL and BeTaL’s applicability for this is beyond the scope of the current work. This paper is focused on designing and evaluating a framework for dynamic benchmarking.
> To acknowledge and further emphasize this promising direction, we have updated the manuscript to explicitly discuss this connection. Specifically, we have:
> - Expanded the conclusion to highlight curriculum learning as a natural and exciting extension of BeTaL.
> - Added clarifications in the limitations section noting that while BeTaL appears structurally compatible with curriculum learning, empirical verification is left for future work.
> - Included a description in the future work section outlining how BeTaL could be explored as a foundation for curriculum-based data selection strategies.
>
> We appreciate the suggestion to articulate this broader impact and believe these revisions improve the clarity and forward-looking perspective of the paper.
>
> > W3 & Q2: Error bounds are very large
>
> Thank you very much for pointing out this gap. We fully agree that reducing variance is important for drawing clearer performance comparisons. Due to limited resources, we were unfortunately unable to rerun all methods with substantially more seeds as suggested.
> However, we recognized the importance of improving statistical reliability wherever possible. In the updated version, we aggregate results across the three independent designer models (GPT-5, Opus-4.1, and GROK-4). Since each method is evaluated with three seeds per designer, this gives us an effective sample size of n = 9 independent runs for every method–task pair.
> As shown in Table 1, this aggregation meaningfully stabilizes the results. BeTaL still consistently achieves the smallest performance gaps across all tasks, and its confidence intervals remain tight, with the largest width only 2.93%. In contrast, BoN-TM and BoN-ML continue to exhibit noticeably higher variance, with confidence intervals typically in the 5–9% range even after averaging over nine runs.
> These findings not only reaffirm BeTaL’s superior performance but also strengthen the evidence for its robustness. We appreciate the reviewer’s suggestion, and while running many more seeds remains computationally prohibitive, the expanded aggregation already provides clearer and more statistically reliable comparisons.

---

> ### Author Response · Authors · 2025-12-03
> **Replies to Reviewer gkRb, Part 2**
>
> > W4 & Q3: Bayesian Optimization Baseline
>
> Thank you for this suggestion. Indeed, Bayesian optimization approaches could serve as a black-box optimizer for similar settings. We focused on baselines that capture similar ideas but are better matched to our constraints (in our case, for example, mixed distributions, constrained parameter spaces, and very small budgets).
>
> Concretely, the RS+PPR baseline plays the role of a simple model-free black-box optimizer, while BoN-ML is a surrogate-based method that trains a predictive model on parameter–performance pairs and then selects configurations with low predicted gap. This is structurally similar to BO, where a surrogate is used in place of our learned regression model.
>
> We chose this formulation for several reasons. First, standard (e.g., Gaussian process-based) BO typically assumes low-dimensional continuous spaces and incurs potentially sizeable overhead per iteration. Second, our design spaces include many discrete and combinatorial choices (these include e.g., subsets of actions or tool combinations), where simple evolutionary / replay-style baselines and learned surrogates are often more robust in practice than off-the-shelf Bayesian optimization.

---

### Official Review · Reviewer_ELKE · 2025-11-02

**Soundness:** 2
**Presentation:** 3
**Contribution:** 2
**Rating:** 4
**Confidence:** 3

**Summary:**

This paper addresses the problem of LLM evaluation, where static benchmarks quickly become saturated and dynamic benchmarks are costly to create and maintain manually. The authors propose BeTaL (Benchmark Tuning with an LLM-in-the-loop), a framework to automate the design of dynamic benchmarks. BeTaL starts with an under-specified environment template defined by a set of parameters. A designer LLM is prompted to reason over this parameter space and propose a specific configuration. This configuration is then used by a simulator to generate a set of problems. A target model (the model being evaluated) attempts these problems, and its performance is measured against a desired target performance level. The resulting performance gap is formatted as natural language feedback and included in the designer LLM's prompt for the next iteration. This loop repeats, and the goal is to output the parameter configuration that achieved the minimum gap.

The authors demonstrate BeTaL on three tasks: Arithmetic Sequences, Spatial Reasoning, and $\tau$-bench Airline. Experiments show that BeTaL more effectively finds parameters that match a target difficulty level compared to baselines like random sampling and best-of-n method. The paper also shows that the difficulty of these generated benchmarks transfers across different evaluation models (e.g., from o4-mini to Gemini 2.5 Flash and Claude 3.7 Sonnet).

**Strengths:**

The paper tackles a clear and significant problem for the community: the saturation of static evaluation benchmarks and the high cost of manually updating dynamic ones. The goal of automating this process is well-motivated and valuable.

**Weaknesses:**

1. The framework's primary weakness is its reliance on access to parameterized and verifiable simulators. This assumption is extremely strong and does not hold for many, if not most, complex and realistic evaluation domains. While feasible for the toy-like "Arithmetic Sequences" or "Spatial Reasoning" grid world, this is a critical bottleneck for applying BeTaL to open-ended domains like agentic web tasks, robotics, or complex code generation, where a verifiable simulator is often as hard to build as the evaluation itself.
2. Following the first point, the experimental validation is confined to highly structured and relatively simple environments. It is not clear how the BeTaL framework would scale to tasks with high-dimensional, continuous, or combinatorially vast parameter spaces. The paper itself notes the evaluation is "limited to a small set of domains, leaving multimodal and more subjective tasks unexplored", but this weakness is central to the method's potential impact.
3. The "Designer" LLM's adaptation is purely based on in-context reasoning, where the history is fed back into the prompt. The designer model itself does not learn or update its parameters to become a better designer over time. This approach is capped by the base LLM's reasoning ability and may be inefficient, as the LLM must re-evaluate the entire history in-context at each step.
4. A crucial part of "automating benchmark design" is defining the parameter space itself. The paper briefly explores this in C4 , but the experiment shows the AI-designed space underperforms the human-designed one for "Easy" and "Trivial" benchmarks. This vital part of the problem feels more like an afterthought than a core component of the method.

**Questions:**

1. How do the authors envision BeTaL being applied to domains where a fast, parameterized, and verifiable simulator is not available? For example, evaluating the difficulty of mathematical proofs or the quality of generated code, where "ground truth" is complex and simulation is not applicable. Does the reliance on a simulator  fundamentally limit the method to toy or game-like environments?
2. BeTaL requires a strong LLM as the designer. Does that mean that the designed benchmarks are only be useful to evaluate weaker models instead of strong model such as in the same intelligence level of the designer?

---

> ### Author Response · Authors · 2025-12-03
> **Replies to Reviewer ELKE**
>
> > W1, W2 & Q1: Reliance on access to parameterized and verifiable simulators, limiting applicability to toyish domains
>
> Thank you for this important question! We fully agree on the importance of extendability to a wider range of environments. Below, we clarify what BeTaL requires to be applied to an environment, why the requirements are not as restrictive as it may appear, and how the method naturally extends to more realistic domains that you highlight, i.e., math, coding.
>
> 1. *BeTaL does not require a verifiable and parametrizable simulator*
> A key misunderstanding is that BeTaL requires a fully verifiable or tightly parameterized simulator. In practice, BeTaL needs only two ingredients: (i) **a high-quality way to generate tasks** and (ii) **a task space that exposes meaningful complexity dimensions.** BeTaL is applicable as long as these two components exist (which is true for most modern evaluation benchmarks.) Moreover, such complexity dimensions do not need to be hard-coded to the task generation process and can also be achieved in implicit ways such as data filtering, as long as the generated benchmarks is controllable by the dimension configurations.
>
> 2. *Existing High Quality Task Generators*
> For (i), we have thankfully observed availability of predefined synthetic generators for high-quality evaluation tasks. Such generators already span a wide range domains that BeTaL could leverage.
> Moreover, recent work such as AutoEnv[9] is beginning to automatically construct realistic environments from seed specifications, further expanding the range of applicable tasks and naturally complementing BeTaL’s design loop. Example generators include:
>     * STEM/Reasoning: BenchMaker[1], SAND-Math[2], MathGenie[3]
>     * Coding: AutoCodeBench, DomainEval[5], EvoCodeBench[6]
>     * Agents / Planning: Simia-Tau[7]
>     * Long-context / multi-hop reasoning: MDBench[8]
>
> 3. *Current Agentic/Reasoning Tasks are Naturally Parametrizable*
> The core idea behind BeTaL is that we only need a high-quality task generator and a set of benchmark parameters that correlate with task complexity. Such parameters exist for almost all tasks, so we don’t require environments to be strongly parameterizable. However, we find a particular fit environment would be meaningfully scalable along two key axes: (1) **the size of the problem space**, and (2) **the length or number of steps** needed to reach the outcome.
> This aligns with most multi-step reasoning and agentic benchmarks. For instance, mathematical-reasoning tasks can vary in the number of reasoning hops or the complexity of operations involved. Similarly, coding tasks can vary with the number of files touched, the number of lines of changes, or the number of libraries involved. Across domains, these two axes provide the basis for BeTaL’s complexity tuning and benchmark construction.
> Despite that the generators we use are verifiable, we believe it is also reasonable and beneficial to extend to many environments **without requiring full verifiability** for cost efficiency. Prior work has repeatedly shown that high-quality benchmarks can be generated without symbolic correctness signals: BenchMaker [1] produces high-quality synthetic math and STEM benchmarks using multi-stage LLM-as-judge pipelines, achieving credibility and diversity on par with human-authored tasks. Similarly, SAND-Math [2] generates strong math datasets through layered filtering and validation steps. In the agentic space, Simia-Tau[7] shows that LLM-formalized rules also supervises complex agent trajectories.
>
> 4. *BeTaL in a Bigger Scope*
> We want to emphasize that BeTaL is an pinioring attempt toward automated, model-adaptive benchmark design. While it does not replace full environment construction, it provides a practical and scalable approach for configuring existing task generators to match specific models and evaluation goals.
>
> [1]https://arxiv.org/html/2502.01683v1
>
> [2]https://arxiv.org/html/2507.20527v3
>
> [3]https://aclanthology.org/2024.acl-long.151.pdf
>
> [4]https://arxiv.org/pdf/2508.09101
>
> [5]https://arxiv.org/pdf/2408.13204
>
> [6]https://arxiv.org/pdf/2404.00599
>
> [7]https://arxiv.org/html/2511.01824v1
>
> [8]https://arxiv.org/pdf/2506.14927
>
> [9]https://arxiv.org/pdf/2511.19304

---

> ### Author Response · Authors · 2025-12-03
> **Replies to Reviewer ELKE, Part 2**
>
> > W3: Designer is Purely Based on Incontext-Learning
>
> Thank you for the thoughtful observation. To clarify, in our implementation the designer LLM does not accumulate the full interaction history; instead, we provide only the most recent parameter setting and performance feedback at each round. This keeps the prompt length fixed and ensures that the cost does not scale with the number of iterations, so the concern about repeatedly prompting with the entire history does not apply in our setting.
>
> We agree that improving the designer to improve over time is an interesting future direction. However, our results show that even this purely in-context approach with constant context is already highly effective: across all environments, designers produce benchmarks whose average performance gaps are within 10.39% of the target at maximum, despite the performance gap can be as large as 90% in our setting. This suggests that modern LLMs are already strong parameter optimizers in designing benchmarks, even without continual improvement or iterative adaptation of their own parameters.
>
> Overall, we believe that more sophisticated designer improvement mechanisms could further enhance performance, yet we demonstrate that a simple, cost-bounded in-context strategy is already a robust and generalizable solution for automated benchmark design.
>
> > W4: AI-Designed Space Underperforms Human-Designed One for "Easy" and "Trivial" benchmarks
>
> Thank you for raising this important point. We fully agree that defining the parameter space is a crucial component of automated benchmark design, which is why we conducted the additional experiment comparing AI-designed and human-designed spaces in Section 4, C4. While the AI-designed space does underperform on the “Trivial” and “Easy” settings (90% and 75% success rates respectively), we find that it performs comparably to the human-designed space on the “Medium” (50%) and “Hard” (25%) settings. Averaged over three designer models, this yields a performance gap of only 7.68%, still outperforming all baselines. This indicates that BeTaL remains effective even when the underlying parameter space is automatically constructed.
>
> The experiment is intended as an exploratory stress test rather than a core component of the main method. Still, it provides a valuable takeaway: while LLMs can meaningfully propose parameter spaces, fully automating this component remains challenging. Based on current results, we recommend a human–AI collaborative workflow, where humans provide coarse task primitives and LLMs explore or refine the parameterization automatically.

---

> ### Author Response · Authors · 2025-12-03
> **Replies to Reviewer ELKE, Part 3**
>
> > Q2: Evalution with model at the same intelligence level of the designer
>
> We also recognize the importance of designing benchmarks that remain meaningful for models with capabilities comparable to the designer model. Due to experimental budget constraints, we were not able to include GPT-5 as a target model for BeTaL in the loop. However, we conducted an exploratory study to assess how well benchmarks designed by GPT-5 for o4-mini transfer to GPT-5 itself when used solely as an evaluation model.
>
> | target performance | Hard (25%)      | Medium (50%)   | Easy (75%)      | Trivial (90%) |
> |--------------------|-----------------|----------------|------------------|----------------|
> | **Spatial Reasoning** | 71.2 ± 4.2      | 99.4 ± 0.4     | 99.8 ± 0.45       | 100 ± 0        |
> | **Tau-Bench**         | 22.7 ± 7.59     | 36 ± 18.76     | 55.3 ± 7.59       | 52.7 ± 20.69   |
> | **Arith. Seq.**       | 43.6 ± 24       | 67.1 ± 43      | 79.6 ± 24.8       | 91.6 ± 24.8    |
>
> **Table 1:** *GPT-5's performance across different target performances on all three environments when evaluated using benchmarks designed for o4-mini with GPT-5 as the designer. Numbers are shown in percentage, and 95% confidence intervals are reported.*
>
> Although the benchmarks were optimized for o4-mini, GPT-5 **does not consistently outperform the designed target**, indicating that the difficulty levels induced by BeTaL generalize beyond the target model in the loop. On Arithmetic Sequences, GPT-5’s accuracy increases smoothly along the intended difficulty tiers, showing that the generated parameters remain meaningful even for substantially stronger models. Tau-Bench performance similarly remains unsaturated and roughly follows the expected ordering. In contrast, Spatial Reasoning is almost fully saturated by GPT-5, indicating that transferability can break down for tasks where the capability gap between target and evaluator is large.
>
> Together, these results offer early but clear evidence of transferability to higher-capability models. Two of the three benchmarks retain their intended difficulty structure when evaluated by GPT-5, and this difficulty structure would likely be even stronger if GPT-5 were used directly as the target model in the loop. Although preliminary, these findings support our broader claim that strong models can design benchmarks that are meaningful for peers with similar intelligence. We plan to continue this line of experimentation after the rebuttal period as resources allow.

---

### Official Review · Reviewer_wqbA · 2025-11-08

**Soundness:** 3
**Presentation:** 3
**Contribution:** 2
**Rating:** 6
**Confidence:** 3

**Summary:**

The authors are responding to an existing need in the LLM community to provide challenging yet realistic evaluation methods to measure the performance of the ever evolving LLMs. The approach is well motivated and the topic of automated LLM benchmarking has been an important discussion point in both academic and industry settings. The authors "formulate the benchmark design process as an optimization problem, where the goal is to maximize utility or usefulness," introducing a "design process for automatically producing and evolving benchmark(s)," to ensure that the ever evolving LLMs can be matched with ever adapting evaluation methods.

**Strengths:**

The authors define the "benchmark design process as an optimization problem" and demonstrate that their "empirical results show BeTaL consistently obtains benchmarks with any given target difficulty, achieving a performance gap of as low as 0.4% and up to 5% in several settings, a significant improvement over baselines."

This is an important contribution, leading to environment setups that are well suited to a specific performance level and can be used to test a variety of different models. Such ability is significant, given that modern LLMs have a wide range of capabilities and differ from one another greatly, hence cannot all be measured by the standard of the best performing competition LLMs, which are often the ones leading to SoTA results on benchmarks.

**Weaknesses:**

While the paper provides a valuable contribution toward a benchmark contains tasks with different complexity levels, I believe to strengthen the authors claims' about its universality and adaptability, the benchmarks should have more been tested on target models of different sizes. It is a bit unclear to me what is the different between the 'target' and 'evaluating' models provided by authors in Section 4.3, stating that 'We use o4-mini as the target model in all the settings. We finally evaluate benchmarks developed by each method on three models: o4-mini, Gemini 2.5 Flash, and Claude 3.7 Sonnet," but all of these models are in the closed source, large scale model category, hence limiting the weight of the authors' claim as to how universal generated experiments are.

I would like to see more details on the efficiency analysis of the proposed algorithm. While Figure 2 shows relative performance gain of BeTaL over the baseline, the authors do not provide any analysis of the convergence rate of the algorithm to generate the target performance level benchmarks vs the search parameter space, with the latter varying greatly between the different experiment setups.

Another necessary information missing is the rate of hallucinations by the model, demonstrating how often BeTaL generates out-of-domain configuration suggestions.

Authors in Conclusion mention “efficient task synthesis,” however later in Limitations directly state that the simulators themselves are assumed to already exist. That makes it unclear whether the authors method, based on Algorithm 1, does not exist to synthesize tasks (aka user environments), but only to optimize them to a target performance level, or whether it can do both.

**Questions:**

Could the authors clarify exactly the scope of the BeTaL algorithm. Is it accurate that its goal is to synthetize new parameters for existing environment to suit a target performance setting, rather than designing completely new ones?

In general, I believe that in order for the BeTaL to be most useful to the community, authors should provide more details about the environment choice step. What considerations should the people who use PeTaL follow to choose an optimal environment, is there an abundance of some predefined environments that the authors can recommend BeTaL users could start with, or is environment design and a setup still essentially a manual process, with BeTaL just meaning to optimize its configurations to a specific target LLM performance level?

---

> ### Author Response · Authors · 2025-12-03
> **Replies to Reviewer wqbA**
>
> Thank you for your thoughtful and constructive feedback, and for recognizing the motivation, adaptability, and contribution to a timely problem of our automated evaluation framework! We address your comments point by point below:
>
> > W1: Unclarity of Target/Eval Model Definition; test on target models of different sizes
>
> Thank you for pointing out this ambiguity, we’ve clarified the two terms in the updated paper.
>
> In BeTaL, the **target model** is the model used inside the iterative design loop: it provides feedback that guides the construction of benchmark parameters, using relatively small rollout budgets. **Evaluation models** are used after the design process to assess the resulting benchmarks’ actual difficulty and generalizability.
>
> We also appreciate the suggestion to evaluate across different model sizes.  In the revised version, we include two open source models of varied sizes (Mistral-small-3.2-24b, Qwen-2.5-72b) as evaluation models for BeTaL-generated benchmarks on Spatial Reasoning and Tau Bench in Figure 4. Due to budget and time constraints, we were not able run the same experiments for Arithmetic Sequence during the rebuttal period, but we plan to complete and include them in the camera-ready version. The added models exhibit difficulty trends **highly consistent** with the three closed-source models, supporting BeTaL’s generalizability across model families and sizes, and showing that it captures genuine cognitive difficulty rather than overfitting to a particular target model.
>
> We have updated Section 4 and Appendix C.2 accordingly.
>
> > W2: Efficiency and Convergence Analysis
>
> Thank you for this insightful point. We have added an appendix with convergence stability analysis addressing your concern about convergence behavior across varying parameter spaces.
>
> Please see Appendix A.5 which provides a rolling standard deviation plot (Figure 8) demonstrating that BeTaL exhibits **more stable and early convergence** than the RS+PPR baseline across all datasets, despite vast differences in parameter space sizes between experimental setups. BeTaL maintains 5-20% standard deviation throughout parameter search, while RS+PPR shows ~25% variability with minimal improvement over time. This stability analysis confirms that BeTaL not only converges faster (as shown in Figure 3) but does so more predictably and reliably across different search spaces.
>
>
> > W3: Hallucination Analysis
>
> Thank you for raising this point! We agree that hallucination rates would further validate BeTaL’s approach for LLM-guided parameter generation.
>
> We re-examine our experiment records and find the following rate of generating out-of-bound parameter sets across all designer models, GPT-5, Opus-4.1, and GROK 4:
> * Arithmetic sequences:  7.41%
> * Spatial reasoning: 5.43%
> * Tau-Bench: 0%
>
> As reported in the submitted paper, BeTaL’s approach for handling out of domain parameters is to support retries and finally, if still out-of-domain, randomly project the proposed parameter onto the support set, thus guaranteeing all parameters are valid through downstream experiments. Due to the relatively small hallucination rate and the robust fallback strategy, we believe hallucination has a minial effect on our experiment results.

---

> ### Author Response · Authors · 2025-12-03
> **Replies to Reviewer wqbA, Part 2**
>
> > W4 & Q1: Scope and Applicability of BeTaL
>
> Thank you for the thoughtful question! We appreciate your suggestion and clarify the scope and the considerations for choosing suitable environments below.
>
> 1. *Scope: BeTaL optimizes existing task generators, not full environment design*
> BeTaL’s primary purpose is to optimize parameters of an existing task generation mechanism (whether manually specified, LLM-generated, or hybrid) to achieve tasks at a target performance level. The algorithm does assume access to some way of generating tasks given parameters.
> Even under this modest assumption, BeTaL addresses a major community gap: today’s evaluation ecosystem still relies heavily on static benchmarks, which are inflexible and rarely aligned with the capabilities of different models or development stages. Enabling **targeted, difficulty-controlled** task synthesis meaningfully improves the adaptability and coverage of evaluations.
>
> 2. *Environment Choice: What kinds of environments work best with BeTaL?*
> The core idea behind BeTaL is that we only need a high-quality task generator and a set of benchmark parameters that correlate with task complexity. Such parameters exist for almost all tasks, so we don’t require environments to be strongly parameterizable. However, we find a particular fit environment would be meaningfully scalable along two key axes: (1) **the size of the problem space**, and (2) **the length or number of steps** needed to reach the outcome.
> This aligns with most multi-step reasoning and agentic benchmarks. For instance, mathematical-reasoning tasks can vary in the number of reasoning hops or the complexity of operations involved. Similarly, coding tasks can vary with the number of files touched, the number of lines of changes, or the number of libraries involved. Across domains, these two axes provide the basis for BeTaL’s complexity tuning and benchmark construction.
> Despite that the generators we use are verifiable, we believe it is also reasonable and beneficial to extend to many environments **without requiring full verifiability** for cost efficiency. Prior work supports this approach: BenchMaker [1] produces high-quality synthetic math and STEM benchmarks using multi-stage LLM-as-judge pipelines, achieving credibility and diversity on par with human-authored tasks. Similarly, SAND-Math [2] generates strong math datasets through layered filtering and validation steps.
>
> 3. *Availability of Predefined Generators*
> In practice, a lot of community works already exists that leverage synthetic generators for high-quality evaluation tasks. Such generators already span a wide range domains that BeTaL could leverage:
>     * STEM/Reasoning: BenchMaker[1], SAND-Math[2], MathGenie[3]
>     * Coding: AutoCodeBench, DomainEval[5], EvoCodeBench[6]
>     * Agents / Planning: Simia-Tau[7]
>     * Long-context / multi-hop reasoning: MDBench[8]<br>
>
>     Moreover, recent work such as AutoEnv[9] is beginning to automatically construct verifiable environments from seed specifications, further reducing the need for manual setup and naturally complementing BeTaL’s design loop.
>
> 4. *BeTaL in a Bigger Scope*
> We want to emphasize that BeTaL is the first systematic attempt toward automated, model-adaptive benchmark design. While it does not replace full environment construction, it provides a practical and scalable approach for configuring existing task generators to match specific models and evaluation goals.
>
>
> [1]https://arxiv.org/html/2502.01683v1
>
> [2]https://arxiv.org/html/2507.20527v3
>
> [3]https://aclanthology.org/2024.acl-long.151.pdf
>
> [4]https://arxiv.org/pdf/2508.09101
>
> [5]https://arxiv.org/pdf/2408.13204
>
> [6]https://arxiv.org/pdf/2404.00599
>
> [7]https://arxiv.org/html/2511.01824v1
>
> [8]https://arxiv.org/pdf/2506.14927
>
> [9]https://arxiv.org/pdf/2511.19304

---

### Author Response · Authors · 2025-12-03
**Summary of Rebuttal Revisions**

Thank you for reviewing our work. We are encouraged that the reviewers recognize this paper as “an important contribution leading to environment setups that are well suited to a specific performance level” (wqbA). We also appreciate the shared sentiment that our goal of automating benchmark design is “well-motivated and valuable” (ELKE) and addresses an “interesting, and an important problem” (gkRb) in combating benchmark saturation, an issue that remains underexplored in the community. We are grateful for the thoughtful suggestions, which have helped us substantially strengthen the revised version.

In response to the comments, we added aggregated results over more independent runs, which significantly tightened confidence intervals and made comparisons across methods clearer. We also expanded our analyses on efficiency, convergence, and hallucination, showing how BeTaL behaves—and improves—through the iterative design process. In addition, we ran new evaluations on both smaller open-source models and state-of-the-art models with similar intelligence to the designers, confirming that BeTaL’s benchmark settings transfer well to both weaker and stronger evaluators.

Finally, we clarified the scope and novelty of the work: BeTaL is the first systematic attempt at automated, model-adaptive benchmark design. It offers a broadly applicable framework that aligns with emerging community trends toward scalable, dynamic evaluation. This is significant as our community moves beyond static, hand-crafted benchmarks and toward continuously evolving testbeds capable of matching the pace of modern LLM development.

---

### Meta-Review · Area_Chair_HXci · 2026-01-04

**Summary:**

The decision to reject is primarily driven by concerns regarding the framework's scope and applicability. While the motivation to automate benchmark design is commendable, reviewers consistently pointed out that the reliance on pre-existing, verifiable, and parameterized simulators constitutes a strong assumption that limits the method's utility to toy-like or highly structured environments. Furthermore, the lack of comparison with standard Bayesian Optimization baselines and the omission of curriculum learning experiments left the empirical validation insufficient to demonstrate broad impact.

**Reviewer Concerns:**

The authors successfully addressed several technical queries, including convergence stability analysis and hallucination rates, and reduced experimental variance by aggregating results across multiple runs. However, critical concerns remain outstanding. Reviewer ELKE's fundamental critique regarding the bottleneck of requiring parameterized simulators was not fully resolved; the rebuttal clarified the scope but confirmed this limitation. Additionally, Reviewer gkRb's concern about the toyish nature of the environments and the lack of standard Bayesian Optimization baselines persists, as the authors argued against the baseline rather than providing a direct empirical comparison.

**Reviewer Scores:**

Reviewer wqbA would likely maintain their score of 6, as their specific technical questions regarding model diversity and efficiency were well-answered. However, Reviewers ELKE and gkRb would likely maintain their scores of 4. The rebuttal clarified the method's constraints but did not fundamentally alter the perception that the framework is limited to optimizing parameters within existing generators rather than solving the broader challenge of benchmark creation, leaving the skeptical reviewers unconvinced of its general applicability.

---

### Decision · Program_Chairs · 2026-01-26

Reject